# Risk-Controlling Model Selection via Guided Bayesian Optimization

**Bracha Laufer-Goldshtein**  *blaufer@tauex.tau.edu*
*Department of Electrical Engineering*
*Tel-Aviv University*

**Adam Fisch**  *fisch@google.com*
*Google DeepMind*

**Regina Barzilay**  *regina@csail.mit.edu*
*Computer Science and Artificial Intelligence Laboratory (CSAIL)*
*Massachusetts Institute of Technology*

**Tommi Jaakkola**  *tommi@csail.mit.edu*
*Computer Science and Artificial Intelligence Laboratory (CSAIL)*
*Massachusetts Institute of Technology*

**Reviewed on OpenReview:** *https://openreview.net/forum?id=nvmGBcElus*

## Abstract

Adjustable hyperparameters of machine learning models typically impact various key trade-offs such as accuracy, fairness, robustness, or inference cost. Our goal in this paper is to find a configuration that adheres to user-specified limits on certain risks while being useful with respect to other conflicting metrics. We solve this by combining Bayesian Optimization (BO) with rigorous risk-controlling procedures, where our core idea is to steer BO towards an efficient testing strategy. Our BO method identifies a set of Pareto optimal configurations residing in a designated region of interest. The resulting candidates are statistically verified, and the best-performing configuration is selected with guaranteed risk levels. We demonstrate the effectiveness of our approach on a range of tasks with multiple desiderata, including low error rates, equitable predictions, handling spurious correlations, managing rate and distortion in generative models, and reducing computational costs.[1]

## 1 Introduction

Deploying machine learning models in the real-world requires balancing different performance aspects such as low error rate, equality in predictive decisions (Hardt et al., 2016; Pessach & Shmueli, 2022), robustness to spurious correlations (Sagawa et al., 2019; Yang et al., 2023), and model efficiency (Laskaridis et al., 2021; Menghani, 2023). In many cases, we can influence the model's behavior favorably via hyperparameters that determine the model configuration. However, selecting a configuration that accurately aligns with user-specified requirements on test data can be particularly challenging. This complexity is amplified when the process involves numerous possible configurations that demand significant resources to evaluate, such as those necessitating the retraining of large neural networks for novel settings.

Bayesian Optimization (BO) is widely used for efficiently selecting configurations of functions that require expensive evaluation, such as hyperparameters that govern the model architecture or influence the training procedure (Shahriari et al., 2015; Wang et al., 2022; Bischl et al., 2023). The basic concept behind BO is to substitute the costly function of interest with a cheap, easily optimized probabilistic surrogate model. This surrogate is then used to select promising candidate configurations while balancing exploration and

---

[1]Our code is available at https://github.com/bracha-laufer/guidebo.

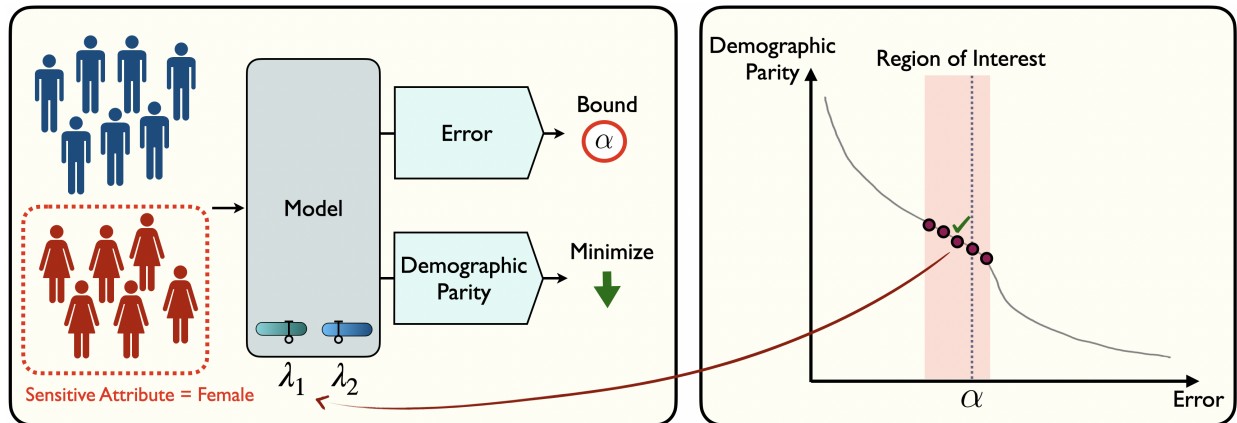

Figure 1: Demonstration of GuideBO for algorithmic fairness with gender as a sensitive attribute (left). We would like to set the model configuration $\boldsymbol{\lambda} = [\lambda_1, \lambda_2]$ to minimize the difference in demographic parity, while bounding the overall prediction error by $\alpha$. Our method (right): (i) defines a region of interest in the objective space, (ii) identifies Pareto optimal solutions in this region, (iii) statistically validates the chosen solutions, and (iv) sets $\boldsymbol{\lambda}$ to the best-performing verified configuration.

exploitation. Beyond single-function optimization, BO has been extended to handle multiple objectives. In this context, the goal is to find a set of Pareto optimal configurations that represent the best possible trade-offs for the given objectives (Karl et al., 2022). Additionally, BO can accommodate multiple inequality constraints (Gardner et al., 2014). Nevertheless, none of these mechanisms provide formal guarantees on model behavior at test time, and can suffer from unexpected fluctuations from the desired final performance (Letham et al., 2019; Feurer et al., 2023).

The Learn then Test (LTT) framework (Angelopoulos et al., 2021) addresses model configuration selection from a different perspective. It establishes a rigorous statistical testing approach for simultaneously controlling multiple risk functions. The procedure is model-agnostic, distribution-free, and yields finite-sample guarentees. While LTT provides exact theoretical verification, its practical application becomes challenging when dealing with large configuration spaces. The increased computational costs and potential loss of statistical power hinder the identification of useful configurations. To mitigate these challenges, the recently proposed *Pareto Testing* method (Laufer-Goldshtein et al., 2023) combines the strengths of multi-objective optimization and statistical testing. The fundamental idea is to leverage multi-objective optimization to significantly reduce the space of potential configurations. This approach aims to identify Pareto optimal configurations that are promising candidates for testing. While this method enhances computational and statistical efficiency, the identified subspace may still include configurations that are irrelevant. It may focus on configurations that are either valid yet inefficient, or that highly improbable to meet the constraints. Therefore, when considering expansive configuration spaces, this strategy can again become costly and statistically loose.

In this work, we introduce GuideBO, a new synergistic approach to combine optimization and testing to achieve efficient model selection under multiple risk constraints. Our approach centers around the concept of the "region of interest" in the objective space, which aligns with the goal of achieving testing efficiency while operating within a limited compute budget. To define the region of interest, we consider factors such as data sample sizes, user-specified limits, and required certainty levels. Consequently, we propose an adjusted BO procedure, recovering the part of the Pareto front that intersects with the defined region of interest. The resulting focused optimization procedure recovers a dense set of configurations, representing candidates that are both effective and likely to pass the test. In the final step, we apply statistical testing to filter this chosen set and identify highly-performing configurations that exhibit verified control.

We demonstrate that GuideBO is a flexible approach applicable across diverse contexts for both predictive and generative models. It effectively tunes various types of hyperparameters that impact the model—whether prior to training or post-training. Specifically, we show its applicability in the domains of algorithmic fairness, robustness to spurious correlations, rate and distortion in Variational Autoencoders (VAEs), accuracy-cost

trade-offs for pruning computations of large-scale Transformer models, and early-time classification in large language models (LLMs). See Fig. 1 for an example and a high-level illustration of GuideBO.

**Contribution.** Our main ideas and results can be summarized as follows:

1. We introduce the region of interest in the objective space, which significantly reduces the search space for candidate configurations, thereby leading to more efficient statistical testing with fewer computations.

2. We define a new BO procedure to identify configurations that are Pareto optimal and lie in the defined region of interest. These configurations are subsequently validated through statistical testing.

3. Our approach facilitates risk-controlled model selection in complex and costly settings that necessitate model retraining or involve extensive configuration spaces. We present a broad range of problems, where our approach can be valuable for valid control and effective optimization of diverse performance aspects, including classification fairness, predictive robustness, generation capabilities, model compression and runtime reduction.

4. Through empirical experiments, we demonstrate that GuideBO selects highly efficient and verified configurations under practical budget constraints, outperforming baselines.

## 2 Related work

**Conformal prediction and risk control.** Conformal prediction is a popular model-agnostic and distribution-free uncertainty estimation framework that returns prediction sets or intervals containing the true value with high probability (Vovk, 2002; Vovk et al., 2015; 2017; Lei et al., 2013; 2018; Gupta et al., 2020; Barber et al., 2021). Coverage validity, provided by standard conformal prediction, has recently been extended to controlling general statistical losses, allowing guarantees in expectation (Angelopoulos et al., 2022) or with user-defined probability (Bates et al., 2021). Our contribution extends the foundational work of Angelopoulos et al. (2021), which addresses the broader scenario of controlling multiple risk functions. This is achieved by selecting an appropriate low-dimensional hyperparameter configuration using multiple hypothesis testing (MHT). Additionally, we draw upon the recently introduced Pareto Testing method (Laufer-Goldshtein et al., 2023) that further improves computational and statistical efficiency by solving a multi-objective optimization (MOO) problem and focusing the testing procedure over the approximated Pareto optimal set. In this paper, we point out that recovering the entire Pareto front is redundant and costly and suggest instead to recover a focused part of the front that is aligned with the purpose of efficient testing. This enables highly-expensive hyperparameter tuning that involves retraining of large models with a limited compute budget.

**Black-box model selection under multiple objectives.** The demand for automating the creation of machine learning pipelines is rapidly increasing. Hyperparameter optimization aims to streamline this process by fine-tuning model configurations with minimal manual intervention. However, this is a challenging black-box optimization problem that often involves trade-offs among factors such as predictive performance (Schmucker et al., 2021), computational time (Wang et al., 2019; Lu et al., 2019), and fairness (Pfisterer et al., 2019; Martinez et al., 2020). In this context, we focus on model selection with multiple objectives, incorporating user-specified statistical constraints on certain losses.

**Bayesian Optimization (BO).** BO is a commonly used sequential model-based optimization technique to efficiently find an optimal configuration for a given black-box objective function (Shahriari et al., 2015; Frazier, 2018; Wang et al., 2022). It can be applied to constrained optimization problems (Gardner et al., 2014) or multi-objective scenarios involving several conflicting objectives (Karl et al., 2022). However, when used in model hyperparamaeter tuning, the objective functions can only be approximated through validation data, resulting in no guarantees on test time performance. To account for that we resort to statistical testing, and utilize the effectiveness of BO to efficiently explore the configuration space and identify promising candidates for testing.

**BO and Conformal Prediction.** Several recent works proposed to integrate conformal prediction into BO in order to improve the optimization process under model misspecification and in the presence of

observation noise (Stanton et al., 2023; Salinas et al., 2023). These works go in a different direction from our approach, guaranteeing coverage over the approximation of the surrogate model, while ours provides validity on configuration selection. Another recent work (Zhang et al., 2023) utilizes online conformal prediction for maintaining a safety violation rate (limiting the fraction of unsafe configurations found during BO), which differs from our provided guarantees and works under the assumption of a Gaussian observation noise.

**Multi-Objective Optimization (MOO).** Simultaneously optimizing multiple black-box objective functions was traditionally performed with evolutionary algorithms, such as NSGA-II (Deb et al., 2002), SMS-EMOA (Emmerich et al., 2005) and MOEA/D (Zhang & Li, 2007). Due to the need for numerous evaluations, evolutionary methods can be costly. Alternatively, BO methods are more sample efficient and can be combined with evolutionary algorithms. Various methods were proposed exploiting different acquisition functions (Knowles, 2006; Belakaria et al., 2019; Paria et al., 2020) and selection mechanisms, encouraging diversity in the objective space (Belakaria et al., 2020) or in the design space (Konakovic Lukovic et al., 2020). The central idea behind our approach is to design a Multi-Objective BO (MOBO) procedure that recovers a small set of valid and efficient configurations. Subsequently, we calibrate this chosen set using MHT (Angelopoulos et al., 2021).

Additional related work is given in Appendix A.

## 3    Problem formulation

Consider an input $X \in \mathcal{X}$ and an associated label $Y \in \mathcal{Y}$ drawn from a joint distribution $p_{XY} \in \mathcal{P}_{XY}$[2]. We learn a model $f_{\boldsymbol{\lambda}} \colon \mathcal{X} \to \mathcal{Y}$, where $\boldsymbol{\lambda} \in \Lambda \subseteq \mathbb{R}^n$ is an $n$-dimensional hyperparameter that determines the model configuration. The model weights are optimized over a training set $\mathcal{D}_{\mathrm{train}}$ by minimizing a given loss function, while the hyperparameter $\boldsymbol{\lambda}$ determines different aspects of the training procedure or the final setting of the model. For example, $\boldsymbol{\lambda}$ can weigh the different components of the training loss function, affect the data on which the model is trained, or specify the final mode of operation in a post-processing procedure.

We wish to select a model configuration $\boldsymbol{\lambda}$ according to different, often conflicting performance aspects, such as low error rate, fairness across different subpopulations and low computational costs. In many practical scenarios, we would like to constrain several of these aspects with pre-specified limits to guarantee a desirable performance in test time. Specifically, we consider a set of objective functions of the form $\ell : \mathcal{P}_{XY} \times \Lambda \to \mathbb{R}$. We assume that there are $c$ constrained objective functions $\ell_1, \ldots., \ell_c$, where $\ell_i(\boldsymbol{\lambda}) = \mathbb{E}_{p_{XY}}[L_i(f_{\boldsymbol{\lambda}}(X), Y; \boldsymbol{\lambda})]$ and $L_i : \mathcal{Y} \times \mathcal{Y} \times \Lambda \to \mathbb{R}$ is a loss function. In addition, there is a free objective function $\ell_{\mathrm{free}}$ defining a single degree of freedom for minimization. The selection of $\boldsymbol{\lambda}$ is carried out based on two disjoint data subsets: (i) a validation set $\mathcal{D}_{\mathrm{val}} = \{X_i, Y_i\}_{i=1}^k$ and (ii) a calibration set $\mathcal{D}_{\mathrm{cal}} = \{X_i, Y_i\}_{i=k+1}^{k+m}$. We will use the validation data to identify a set of candidate configurations, and the calibration data to calibrate the identified set. The constraints are formulated by an $(\alpha, \delta)$-risk control criterion as was defined by Angelopoulos et al. (2021):

$$\mathbb{P}_{\mathcal{D}_{\mathrm{cal}}} \left( \ell_i(\boldsymbol{\lambda}) \leq \alpha_i \right) \geq 1 - \delta, \ \ \forall i \in \{1, \ldots, c\}, \tag{1}$$

where $\alpha_i$ is the upper bound of the $i$-th objective function, and $\delta$ is the desired confidence level, both selcted by the user. Note that the probability in (1) is defined over the randomness of the calibration data $\mathcal{D}_{\mathrm{cal}}$, namely if $\delta = 0.1$, then the selected configuration will satisfy the constraints at least 90% of the time across different calibration datasets.

We provide here a brief example of our setup in the context of algorithmic fairness and derive additional applications in §6. In many cases, we wish to increase the fairness of the model without significantly sacrificing performance. For example, we would like to encourage similar true positive rates across different subpopulations, while constraining the expected error. One approach to enhancing fairness involves adding fairness-promoting terms to the standard cross-entropy loss function (Lohaus et al., 2020; Padh et al., 2021; Chuang & Mroueh, 2020). In this case, $\boldsymbol{\lambda}$ contains the weights assigned to each loss term, which are then combined to form the overall training loss. Adjusting these weights leads to different accuracy-fairness trade-offs for the resulting model. Our goal is to select a configuration $\boldsymbol{\lambda}$ that optimizes fairness, while guaranteeing that the overall error would not exceed a certain limit with high probability.

---

[2]We also address unsupervised learning problems, which do not involve labels. For generality, our problem formulation uses both $X$ and $Y$.

## 4    Background

In our method, two critical components play a central role: optimization of multiple objectives and statistical testing for configuration selection. We hereby provide a short overview on these topics.

**Multi-Objective Optimization (MOO).** Consider an optimization problem over a vector-valued function $\boldsymbol{\ell}(\boldsymbol{\lambda}) = (\ell_1(\boldsymbol{\lambda}), \ldots, \ell_d(\boldsymbol{\lambda}))$ consisting of $d$ objectives. When dealing with conflicting objectives, there is no single optimal solution that simultaneously minimizes all objectives. Instead, there is a set of optimal configurations representing different trade-offs among the given objectives. This is the *Pareto optimal set*, defined by:

$$\Lambda_{\mathrm{p}} = \{\boldsymbol{\lambda} \in \Lambda : \{\boldsymbol{\lambda}' \in \Lambda : \boldsymbol{\lambda}' \prec \boldsymbol{\lambda}, \boldsymbol{\lambda}' \neq \boldsymbol{\lambda}\} = \emptyset\}, \tag{2}$$

where $\boldsymbol{\lambda}' \prec \boldsymbol{\lambda}$ denotes that $\boldsymbol{\lambda}'$ *dominates* $\boldsymbol{\lambda}$ if for every $i \in \{1, \ldots d\}$, $\ell_i(\boldsymbol{\lambda}') \leq \ell_i(\boldsymbol{\lambda})$, and for some $i \in \{1, \ldots d\}$, $\ell_i(\boldsymbol{\lambda}') < \ell_i(\boldsymbol{\lambda})$. Accordingly, the Pareto optimal set consists of all points that are not dominated by any point within $\Lambda$. Given an approximated Pareto front $\hat{\mathcal{P}}$, a common quality measure is the hypervolume indicator (Zitzler & Thiele, 1998) defined with respect to a *reference point* $\mathbf{r} \in \mathbb{R}^d$:

$$HV(\hat{\mathcal{P}}; \mathbf{r}) = \int_{\mathbb{R}^d} \mathbb{1}_{H(\hat{\mathcal{P}};\mathbf{r})} d\mathbf{z}, \tag{3}$$

where $H(\hat{\mathcal{P}}; \mathbf{r}) = \{\mathbf{z} \in \mathbb{R}^d : \exists \, \boldsymbol{p} \in \hat{\mathcal{P}} : \mathbf{p} \prec \mathbf{z} \prec \boldsymbol{r}\}$ and $\mathbb{1}_{H(\hat{\mathcal{P}},\mathbf{r})}$ is the Dirac delta function that equals 1 if $\mathbf{z} \in H(\hat{\mathcal{P}}; \mathbf{r})$ and 0 otherwise. An illustration is provided in Fig. B.1. The reference point defines the boundaries for the hypervolume computation. It is usually set to the nadir point that is defined by the worst objective values, so that all Pareto optimal solutions have positive hypervolume contributions (Ishibuchi et al., 2018). For example, in model compression with error and cost as objectives, the reference point can be set to $(1.0, 1.0)$, since the maximum error and the maximum normalized cost equal $1.0$. The hypervolume indicator measures both the individual contribution of each solution to the overall volume, and the global diversity, reflecting how well the solutions are distributed. It can be used to evaluate the contribution of a new point to the current Pareto front approximation, defined as the Hypervolume Improvement (HVI):

$$HVI(\boldsymbol{\ell}(\boldsymbol{\lambda}), \hat{\mathcal{P}}; \mathbf{r}) = HV(\boldsymbol{\ell}(\boldsymbol{\lambda}) \cup \hat{\mathcal{P}}; \mathbf{r}) - HV(\hat{\mathcal{P}}; \mathbf{r}). \tag{4}$$

where $\boldsymbol{\ell}(\boldsymbol{\lambda}) \cup \hat{\mathcal{P}}$ denotes the extended set consisting of the current approximated Pareto front and a new point $\boldsymbol{\ell}(\boldsymbol{\lambda})$. The expression in (4) represents the increase in hypervolume achieved by adding a new point $\boldsymbol{\ell}(\boldsymbol{\lambda})$. The hypervolume indicator serves both as a performance measure for comparing different algorithms and as a score for maximization in various MOO methods (Emmerich et al., 2005; 2006; Bader & Zitzler, 2011; Daulton et al., 2021).

**BO.** BO is an effective method for optimizing black-box objective functions that are costly to evaluate (e.g. selecting the hyperparameters of large neural networks). These methods employ a *surrogate model* to approximate the expensive objective function and iteratively select new configurations using an *acquisition function* that balances exploration and exploitation. Formally, we assume a total budget of $N$ iterations, representing the maximum number of allowed function evaluations. We start with an initial set of random configurations $\mathcal{C}_0 = \{\boldsymbol{\lambda}_0, \ldots, \boldsymbol{\lambda}_{N_0}\}$ and their associated objective values $\mathcal{L}_0 = \{\ell(\boldsymbol{\lambda}_1), \ldots, \ell(\boldsymbol{\lambda}_{N_0})\}$, where $N_0 < N$. We then perform $N - N_0$ iterations, each time adding one configuration to the current set $\mathcal{C}_n$, $1 \leq n \leq N - N_0$, where $|\mathcal{C}_n|$ denotes the set size. Commonly, a Gaussian Process (GP) (Williams & Rasmussen, 2006) serves as a surrogate model, providing an estimate with uncertainty given by the Gaussian posterior. We assume a zero-mean GP prior $g(\boldsymbol{\lambda}) \sim \mathcal{N}(0, k(\boldsymbol{\lambda}, \boldsymbol{\lambda}))$, characterized by a kernel function $\kappa : \Lambda \times \Lambda \rightarrow \mathbb{R}$. Let $\boldsymbol{\kappa} \in \mathbb{R}^{|\mathcal{C}_n|}$ denote the vector of covariance values $\kappa_i = \kappa(\boldsymbol{\lambda}, \boldsymbol{\lambda}_i)$ between a new point $\boldsymbol{\lambda}$ and a given point $\boldsymbol{\lambda}_j \in \mathcal{C}_n$, and $\mathbf{K}$ a $|\mathcal{C}_n| \times |\mathcal{C}_n|$ matrix with elements $K_{ij} = \kappa(\boldsymbol{\lambda}_i, \boldsymbol{\lambda}_j)$, $i, j \in \{1, \ldots, |\mathcal{C}_n|\}$. In addition, we define the label vector $\mathbf{q} \in \mathbb{R}^{|\mathcal{C}_n|}$ consisting of the given objective values $q_i = \ell(\boldsymbol{\lambda}_i), i \in \{1, \ldots, |\mathcal{C}_n|\}$. Based on these definitions, the posterior distribution of the GP is given by $p(g|\boldsymbol{\lambda}, \mathcal{C}_n, \mathcal{L}_n) = \mathcal{N}(\mu(\boldsymbol{\lambda}), \Sigma(\boldsymbol{\lambda}, \boldsymbol{\lambda}))$, with $\mu(\boldsymbol{\lambda}) = \boldsymbol{\kappa}(\boldsymbol{K} + \sigma^2 \mathbf{I})^{-1}\mathbf{q}$ and $\Sigma(\boldsymbol{\lambda}, \boldsymbol{\lambda}) = \kappa(\boldsymbol{\lambda}, \boldsymbol{\lambda}) - \boldsymbol{\kappa}^T (\mathbf{K} + \sigma^2 \mathbf{I})^{-1} \boldsymbol{\kappa}$. Here $\sigma^2$ is the observation noise variance, i.e. $\ell(\boldsymbol{\lambda}_i) \sim \mathcal{N}(g(\boldsymbol{\lambda}_i), \sigma^2)$. The next configuration for evaluation, is selected by optimizing an acquisition function defined on top of the surrogate model. Common acquisition functions include: probability of improvement (PI) (Kushner, 1964), expected improvement (EI) (Močkus,

1975), and lower confidence bound (LCB) (Auer, 2002). For MOO, a GP is fitted to each objective. Then, one approach is to perform scalarization (Knowles, 2006), which converts the problem back into a single-objective optimization, allowing the use of standard acquisition functions. Alternatively, one can employ modified acquisition functions designed for the multi-objective context, such as expected hypervolume improvement (EHVI) (Emmerich et al., 2006) and predictive entropy search for multi-objective optimization (PESMO) (Hernández-Lobato et al., 2016). After selecting a new configuration, it is evaluated and added to the updated set $\mathcal{C}_{t+1}$. This process is repeated until the maximum number of iterations is reached.

**Learn then Test (LTT) & Pareto Testing.** Angelopoulos et al. (2021) have introduced LTT, which is a statistical framework for configuration selection based on multiple hypothesis testing (MHT). Given a set of constraints of the form of Eq. (1), a null hypothesis is defined as $H_{\boldsymbol{\lambda}} : \exists\, i$ where $\ell_i(\boldsymbol{\lambda}) > \alpha_i$ i.e., that at least one of the constraints is *not* satisfied. For a given configuration, we can compute the p-value under the null-hypothesis based on the calibration data. If the p-value is lower than the significance level $\delta$, the null hypothesis is rejected and the configuration is declared to be valid. When testing multiple model configurations simultaneously, this becomes an MHT problem. In this case, it is necessary to apply a correction procedure to control the family-wise error rate (FWER), i.e. to ensure that the probability of one or more wrong rejections is bounded by $\delta$. In large configuration spaces, this can be computationally demanding and result in inefficient testing. In order to mitigate these challenges, Pareto Testing was proposed (Laufer-Goldshtein et al., 2023), where the testing is focused on the most promising configurations identified using MOO. Accordingly, only Pareto optimal configurations are considered and are ranked by their approximated p-values from low to high risk. Then, Fixed Sequence Testing (FST) (Holm, 1979) is applied over the ordered set, sequentially testing the configurations with a fixed threshold $\delta$ until failing to reject for the first time. Although Pareto Testing demonstrates enhanced testing efficiency, it recovers the entire Pareto front, albeit focusing only on a small portion of it during testing. Consequently, the optimization budget is not directly utilized in a way that enhances testing efficiency, putting an emphasis on irrelevant configurations on one side and facing an excessive sparsity within the relevant area on the other, as illustrated in Fig. 2.

## 5 Method

Our approach involves two main steps: (i) performing BO to generate a small set of potential configurations, and (ii) applying MHT over the candidate set to identify valid configurations. Considering the shortcomings of Pareto Testing, we argue that the two disjoint stages of optimization followed by testing are suboptimal, especially for resource-intensive MOO. As an alternative, we propose adjusting the optimization procedure for better testing outcomes by focusing only on the most relevant parts in the objective space. To accomplish this, we need to (i) specify a *region of interest* guided by our testing goal, and (ii) establish a BO procedure capable of effectively identifying configurations within the defined region. In the following we describe these steps in details.

### 5.1 Defining the Region of Interest

We aim to define a region of interest within the objective space $\mathbb{R}^{c+1}$, where we seek to identify candidate configurations that are likely to be valid and efficient during the process of MHT. We start with the case of a single constraint ($c = 1$). Recall that in the testing stage we define the null hypothesis $H_{\boldsymbol{\lambda}} : \ell(\boldsymbol{\lambda}) > \alpha$ for a candidate configuration $\boldsymbol{\lambda}$, and compute a p-value for a given empirical loss over the calibration data $\hat{\ell}^{\mathrm{cal}}(\boldsymbol{\lambda}) = \frac{1}{m} \sum_{j=k+1}^{k+m} \ell(X_j, Y_j; \boldsymbol{\lambda})$. A valid p-value $p_{\boldsymbol{\lambda}}$ has to be super-uniform under the null hypothesis, i.e. $\mathbb{P}\left(p_{\boldsymbol{\lambda}} \leq u\right) \leq u$, for all $u \in [0, 1]$. As demonstrated in (Angelopoulos et al., 2021), a valid p-value can be computed based on concentration inequalities that quantify the proximity of the sample loss to the expected population loss. If the loss is bounded by 1, we can apply Hoeffding's inequality to derive the following p-value (see Appendix B.1):

$$p_{\boldsymbol{\lambda}}^{\mathrm{HF}} := e^{-2m\left(\alpha - \hat{\ell}^{\mathrm{cal}}(\boldsymbol{\lambda})\right)_+^2}. \tag{5}$$

where $(\cdot)_+ = \max(\cdot, 0)$. For a given significance level $\delta$, the null hypothesis is rejected (the configuration is declared to be risk-controlling), when $p_{\boldsymbol{\lambda}}^{\mathrm{HF}} < \delta$. By rearranging (5), we obtain that the maximum empirical

loss $\hat{\ell}(\boldsymbol{\lambda})$ that can pass the test with significance level $\delta$ is given by (see Appendix B.1):

$$\alpha^{\mathrm{max}} = \alpha - \sqrt{\frac{\log{(1/\delta)}}{2m}}. \tag{6}$$

As an example, consider the error rate as a loss function, and assume that we would like to bound the error rate by 5% ($\alpha = 0.05$), with a significance level of $\delta = 0.1$. By (6), if the empirical loss of a calibration set of size $m = 5000$ is up to $\alpha^{\mathrm{max}} = 4\%$, then we have enough evidence to declare that this configuration is safe and its error will not exceed 5% on new unseen data drawn from the same distribution.

In the BO procedure, we are interested in identifying configurations that are likely to be both valid and efficient. On the one hand, in order to be valid the loss must not exceed $\alpha^{\mathrm{max}}$. On the other hand, from efficiency considerations, we would like to minimize the free objective as much as possible. This means that the constrained loss should be close to $\alpha^{\mathrm{max}}$ (from bellow) due to the inverse relation between the free objective and the constrained objective. An illustration demonstrating this idea is provided in Fig. 2, where the irrelevant regions are: (i) the brown part on the right where the configurations are not satisfying the constraint, and (ii) the green part on the left where the configurations are not effectively minimizing $\ell_2$. Ideally, we would like to find configurations with expected loss equal to the limiting testing threshold $\alpha^{\mathrm{max}}$. However, during optimization we can only evaluate the loss over a finite-size validation data with $|\mathcal{D}_{\mathrm{val}}| = k$ samples. To account for that, we construct an interval $[\ell^{\mathrm{low}}, \ell^{\mathrm{high}}]$ around $\alpha^{\mathrm{max}}$ based on the size of the validation data. In this region, we wish to include empirical loss values that are *likely* to correspond to an expected value of $\alpha^{\mathrm{max}}$ based on the evidence provided by the validation data. Let $\hat{\ell}_1^{\mathrm{opt}}(\boldsymbol{\lambda}) = \frac{1}{k} \sum_{j=1}^{k} \ell_1(X_j, Y_j; \boldsymbol{\lambda})$ denote the empirical loss computed over $\mathcal{D}_{\mathrm{val}}$. We define the region $R(\alpha, k, m, \delta, \gamma)$ containing $\hat{\ell}_1^{\mathrm{opt}}(\boldsymbol{\lambda})$ values that are likely to be obtained under $\ell_1(\boldsymbol{\lambda}) = \alpha^{\mathrm{max}}$ with at least $1 - 2\gamma$ probability. Specifically, there exists a region $R(\alpha, k, m, \delta, \gamma)$, with $\gamma \in (0, 0.5]$, such that:

$$\mathbb{P}\left(\hat{\ell}_1^{\mathrm{opt}}(\boldsymbol{\lambda}) \in R(\alpha, k, m, \delta, \gamma) \big| \ell_1(\boldsymbol{\lambda}) = \alpha^{\mathrm{max}}\right) \geq 1 - 2\gamma. \tag{7}$$

For instance, by applying Hoeffding's inequality, we can derive the following region of interest:

$$R(\alpha, k, m, \delta, \gamma) = \left[\underbrace{\alpha^{\mathrm{max}} - \sqrt{\frac{\log{(1/\gamma)}}{2k}}}_{\ell^{\mathrm{low}}}, \underbrace{\alpha^{\mathrm{max}} + \sqrt{\frac{\log{(1/\gamma)}}{2k}}}_{\ell^{\mathrm{high}}}\right]. \tag{8}$$

Note that setting $\gamma$ is an empirical decision that is independent of both the MHT procedure and the chosen significance level $\delta$. For small $\gamma$ the region expands, accommodating more optional configurations but with a lower density. Conversely, a larger $\gamma$ produces a smaller region, leading to denser sampling around the limiting value. Also note that, whenever $k$ increases, the width of the region decreases, reflecting a growing confidence that the observed losses are representative of actual expected loss. In practice, we use the tighter Hoeffding-Bentkus inequality for the p-value computation in Eq. (5) and for defining the region of interest by Eqs. (6) and (8) (see Appendix B.1).

In the case of multiple constraints, the null hypothesis is defined as $H_{\boldsymbol{\lambda}} : \exists\, i$ where $\ell_i(\boldsymbol{\lambda}) > \alpha_i$, i.e. that at least one of the constraints is not satisfied. A valid p-value is given by $p_{\boldsymbol{\lambda}} = \max_{i \in \{1, \dots, c\}} p_{\boldsymbol{\lambda}, i}$, where $p_{\boldsymbol{\lambda}, i}$ is the p-value corresponding to the $i$-th constraint (see Appendix. B.2). Consequently, we define the region of interest in the multi-constraint case as the intersection of the individual regions (as illustrated in Fig. B.2):

$$R(\boldsymbol{\alpha}, k, m, \delta, \gamma) = \bigcap_{i=1}^{c} R(\alpha_i, k, m, \delta, \gamma); \quad \boldsymbol{\alpha} = (\alpha_1, \dots, \alpha_c) \tag{9}$$

## 5.2 Local Hypervolume Improvement

Given our definition of the region of interest, we derive a BO procedure that recovers Pareto optimal points in the intersection of $R(\boldsymbol{\alpha}, k, m, \delta, \gamma)$ and $\mathcal{P}$. Our key idea is to use the HVI in Eq. (4) as an acquisition function, while modifying it to capture only the region of interest. To this end, we properly define the reference point $\mathbf{r} \in \mathbb{R}^{c+1}$ to enclose the desired region.

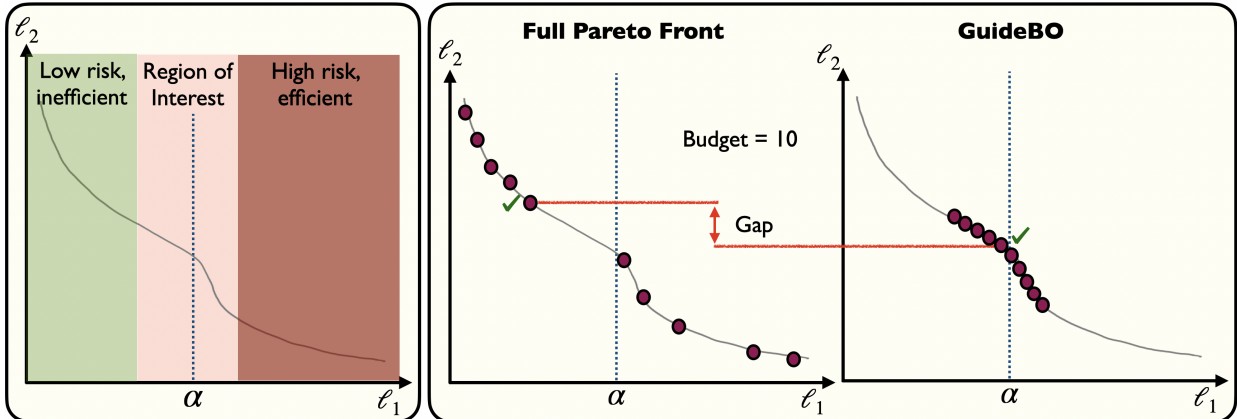

Figure 2: Left: Illustration of the different parts of the Pareto front. The green region consists of configurations that are low risk ($\ell_1 \ll \alpha$) but inefficient in terms of the free objective $\ell_2$. The brown region consists of configurations that are efficient but high risk ($\ell_1 \gg \alpha$) and cannot pass the test. In the middle, we define the region of interest containing configurations that are likely to be both valid and efficient. Right: comparing GuideBO to full Pareto front recovery for optimization budget $N = 10$. In the full Pareto front method there is no control on the distribution of the configurations over the front, while GuideBO focuses on the region of interest. As a result, comparing the chosen valid configurations (marked by v), there exists a noticeable advantage in favor of GuideBO in minimizing $\ell_2$.

Recall that the reference point defines the upper limit in each direction. Therefore, for the constrained dimensions we set $r_i = \ell_i^{\text{high}}$, $i \in \{1, \ldots, c\}$ using the upper bound in Eq. (8). As for the unconstrained dimension $r_{c+1}$, we can use the maximum possible value of $\ell_{\text{free}}$. However, this approach may unnecessarily expand the defined region to include the green area shown in Fig. 2. In this region, the configurations meet the constraint but are sub-optimal with respect to the free objective. Instead, we determine the final dimension based on the lower limiting values. Specifically, we set $r_{c+1}$ to the point on the free axis that corresponds to the intersection of the lower limits of the constrained dimensions:

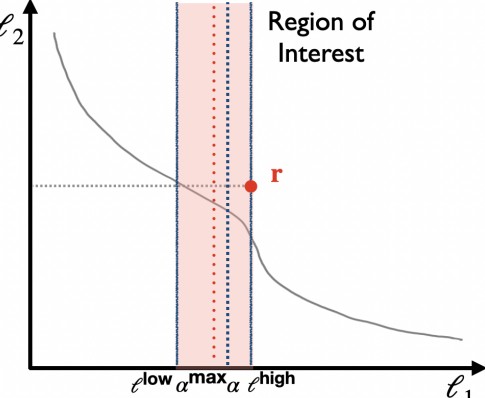

Figure 3: GuideBO for two objectives. $\ell_1$ is controlled at $\alpha$ while $\ell_2$ is minimized. The shaded area corresponds to our defined region of interest. A reference point (in red) is defined accordingly to enclose the region of interest.

$$r_{c+1} = \hat{g}_{\text{free}}(\boldsymbol{\lambda}_{\text{free}}) \tag{10}$$

where

$$\boldsymbol{\lambda}_{\text{free}} = \arg\min_{\boldsymbol{\lambda}} \left\| [\hat{g}_1(\boldsymbol{\lambda}), \ldots, \hat{g}_c(\boldsymbol{\lambda})] - [\ell_1^{\text{low}}, \ldots, \ell_c^{\text{low}}] \right\|_2 \tag{11}$$

where we use the GP posterior mean as our objective estimator, i.e. $\hat{g} = \mu$. As a result, we obtain the following reference point:

$$\mathbf{r} = \left( \ell_1^{\text{high}}, \ldots, \ell_c^{\text{high}}, \hat{g}_{\text{free}}(\boldsymbol{\lambda}_{\text{free}}) \right). \tag{12}$$

We select the next configuration by maximizing the HVI (4) with respect to this reference point:

$$\boldsymbol{\lambda}_n = \arg\max_{\boldsymbol{\lambda}} HVI(\hat{\boldsymbol{g}}(\boldsymbol{\lambda}), \hat{\mathcal{P}}; \mathbf{r}), \tag{13}$$

which leads to recovering only the relevant section, rather than the entire front. We evaluate the objective functions on the new selected configuration, and update our candidate set accordingly. The BO process continues iterating until the maximum budget $N$ is reached. The resulting set of candidate configurations is denoted as $\mathcal{C}^{BO}$. Our proposed BO method, GuideBO, is summarized in Algorithm 1 and illustrated in Fig. 3 for $c = 1$.

---

**Algorithm 1** GuideBO: Testing Guided Bayesian Optimization

---

**Definitions:** $\ell_1, \ldots, \ell_c$ and $\ell_{\text{free}}$ are the objective functions, $g_1, \ldots, g_c$ and $g_{\text{free}}$ are their associated surrogate models. $\ell_1^{\text{low}}, \ldots, \ell_c^{\text{low}}$ and $\ell_1^{\text{high}}, \ldots, \ell_c^{\text{high}}$ are the lower and upper bounds, respectively, for the first $c$ objectives. $\mathcal{C}_0 = \{\boldsymbol{\lambda}_0, \ldots, \boldsymbol{\lambda}_{N_0}\}$ is an initial pool of configurations and $\mathcal{L}_0 = \{\boldsymbol{\ell}(\boldsymbol{\lambda}_1), \ldots, \boldsymbol{\ell}(\boldsymbol{\lambda}_{N_0})\}$ are the associated objectives. $N$ is our total budget. ParetoFront() filters Pareto optimal objective values.

1: **function** BO($\boldsymbol{\ell}, \mathcal{C}_0, \mathcal{L}_0, \{\ell_1^{\text{low}}, \ldots, \ell_c^{\text{low}}\}, \{\ell_1^{\text{high}}, \ldots, \ell_c^{\text{high}}\}, N$)
2:      $N_{\max} \leftarrow N - |\mathcal{C}_0|$
3:      $\mathbf{r} \leftarrow \left( \ell_1^{\text{high}}, \ldots, \ell_c^{\text{high}}, \max_{\boldsymbol{\lambda} \in \mathcal{C}_0} \ell_{\text{free}}(\boldsymbol{\lambda}) \right)$          ▷ Initialize reference point.
4:      **for** $n = 0, 1, 2, \ldots, N_{\max} - 1$ **do**
5:          Fit $\hat{\boldsymbol{g}}$ on $(\mathcal{C}_n, \mathcal{L}_n)$          ▷ Fit surrogate models.
6:          $r_{c+1} \leftarrow \hat{g}_{\text{free}}(\boldsymbol{\lambda}_{\text{free}})$, $\boldsymbol{\lambda}_{\text{free}} = \arg\min_{\boldsymbol{\lambda}} \left\| [\hat{g}_1(\boldsymbol{\lambda}), \ldots, \hat{g}_c(\boldsymbol{\lambda})] - [\ell_1^{\text{low}}, \ldots, \ell_c^{\text{low}}] \right\|_2$      ▷ Update ref. point.
7:          $\hat{\mathcal{P}} \leftarrow \text{ParetoFront}(\mathcal{L}_n)$          ▷ Filter Pareto front according to Eq. (2).
8:          $\boldsymbol{\lambda}_{n+1} = \arg\max_{\boldsymbol{\lambda}} HVI(\hat{\boldsymbol{g}}(\boldsymbol{\lambda}), \hat{\mathcal{P}}; \mathbf{r})$.          ▷ Optimize acquisition function.
9:          Evaluate $\boldsymbol{\ell}(\boldsymbol{\lambda}_{n+1})$          ▷ Evaluate new configuration.
10:         $\mathcal{C}_{n+1} \leftarrow \mathcal{C}_n \cup \boldsymbol{\lambda}_{n+1}$.          ▷ Add new configuration.
11:         $\mathcal{L}_{n+1} \leftarrow \mathcal{L}_n \cup \boldsymbol{\ell}(\boldsymbol{\lambda}_{n+1})$.          ▷ Add new objective values.
12:      $\mathcal{C}^{\text{BO}} \leftarrow \mathcal{C}_{N_{\max}}$
13:      **return** $\mathcal{C}^{\text{BO}}$

---

Note that in MOBO it is common to use an HVI-based acquisition function that also takes into account the predictive uncertainty as in EHVI (Emmerich et al., 2005) and SMS-EGO (Ponweiser et al., 2008). However, our preliminary runs showed that these approaches do not work well in the examined scenarios with small budget ($N \in [10, 50]$), as they often generated points outside the region of interest. Similarly, for these scenarios the random scalarization approach, proposed in (Paria et al., 2020), was less effective for generating well-distributed points inside the desired region.

### 5.3 Testing the Final Selection

We follow (Angelopoulos et al., 2021; Laufer-Goldshtein et al., 2023) for testing the selected candidate set. Prior to testing we filter and order the configurations in the set $\mathcal{C}^{\text{BO}}$. Specifically, we retain only Pareto optimal configurations from $\mathcal{C}^{\text{BO}}$, and then sort the remaining configurations by increasing p-values, approximated by $\mathcal{D}_{\text{val}}$. Next, we recompute the p-values based on $\mathcal{D}_{\text{cal}}$ and perform FST, where we start testing from the first configuration and continue until the first time the p-value exceeds $\delta$. As a result, we obtain the validated set $\mathcal{C}^{\text{valid}}$, and choose a configuration minimizing the free objective:

$$\boldsymbol{\lambda}^* = \arg\min_{\boldsymbol{\lambda} \in \mathcal{C}^{\text{valid}}} \ell_{\text{free}}(\boldsymbol{\lambda}). \tag{14}$$

The method is summarized in Algorithm C.1. As a consequence of (Angelopoulos et al., 2021; Laufer-Goldshtein et al., 2023) we achieve a valid risk-controlling configuration, as we now formally state.

**Theorem 5.1.** *Let $\mathcal{D}_{\text{val}} = \{X_i, Y_i\}_{i=1}^{k}$ and $\mathcal{D}_{\text{cal}} = \{X_i, Y_i\}_{i=k+1}^{k+m}$ be two disjoint datasets. Suppose the p-value $p_{\boldsymbol{\lambda}}$, derived from $\mathcal{D}_{\text{cal}}$, is super-uniform under $\mathcal{H}_{\boldsymbol{\lambda}}$ for all $\boldsymbol{\lambda}$. Then the output $\boldsymbol{\lambda}^*$ of Algorithm C.1 satisfies Eq. (1).*

The proof is provided in Appendix B.3. Note that in situations where we are unable to identify any statistically valid configuration (i.e., $\mathcal{C}^{\text{valid}} = \emptyset$), we set $\boldsymbol{\lambda} = \texttt{null}$. To avoid this situation, the user should select limits $\alpha_1, \ldots, \alpha_c$ that are likely to be feasible. In practice, this can be achieved using the initial pool of configurations $\mathcal{C}_0$, which is generated at the start of the BO procedure. This representative set provides an indication of the possible achievable limits. Specifically, the user may select $\alpha_i \in [\min_{\boldsymbol{\lambda} \in \mathcal{C}_0} \ell_i(\boldsymbol{\lambda}), \max_{\boldsymbol{\lambda} \in \mathcal{C}_0} \ell_i(\boldsymbol{\lambda})], i \in \{1, \ldots, c\}$, and can further refine this choice during the BO iterations as more function evaluations are accumulated.

# 6 Applications

We demonstrate the effectiveness of GuideBO across various tasks with diverse objectives. In each setting, the definition of $\boldsymbol{\lambda}$ varies, affecting the model differently either during or after training.

**Classification Fairness.** In many classification tasks, it is important to take into account the behavior of the predictor with respect to different subpopulations. Assuming a binary classification task and a binary sensitive attribute $a = \{-1, 1\}$, we consider the Difference of Demographic Parity (DDP) as a fairness score (Wu et al., 2019):

$$\text{DDP}(f) = \mathbb{E}\left[\mathbb{1}_{f(x)>0}|a = -1\right] - \mathbb{E}\left[\mathbb{1}_{f(x)>0}|a = 1\right]. \tag{15}$$

We define the following loss, parameterized by $\boldsymbol{\lambda} = [\lambda_1, \lambda_2]$, which consists of two regularization terms that prompt fairness:

$$R(f; \boldsymbol{\lambda}) = \text{BCE}(f) + \lambda_1 \cdot \widehat{\text{DDP}}(f) + \lambda_2 \cdot \widehat{\text{MixUP}}(f), \tag{16}$$

where $\text{BCE}(f)$ is the standard binary cross-entropy loss, and $\widehat{\text{DDP}}(f)$ and $\widehat{\text{MixUP}}(f)$ regularize the model's fairness. $\widehat{\text{DDP}}(f)$ is the hyperbolic tangent relaxation of (15) (Padh et al., 2021), and $\widehat{\text{MixUP}}(f)$ is a mixup regularization interpolating samples between groups (Chuang & Mroueh, 2020) (see Appendix D for further details). Changing the values of $\boldsymbol{\lambda}$ leads to different models that trade-off accuracy for fairness. In this setup, we have a 2-dimensional hyperparamter $\boldsymbol{\lambda}$ and two objectives: (i) the error of the model $\ell_{\text{err}}(\boldsymbol{\lambda}) = \mathbb{E}\left[\mathbb{1}_{f_{\boldsymbol{\lambda}}(X)\neq Y}\right]$, and (ii) the DDP defined in Eq. (15) $\ell_{\text{ddp}}(\boldsymbol{\lambda}) = \text{DDP}(f_{\boldsymbol{\lambda}})$.

**Classification Robustness.** Predictors often rely on spurious correlations found in the data (such as background features), which leads to significant performance variations among different subgroups. Recently, Izmailov et al. (2022) demonstrated that models trained using expected risk minimization surprisingly learn core features in addition to spurious ones. Accordingly, they proposed to enhance model robustness by retraining the final layer on a balanced dataset. We modify their approach to generate different configurations, balancing the trade-off between robustness to differences in subpopulations and overall performance across the entire population.

Given a dataset $\mathcal{D}$ (either the training set or a part of the validation set), we define a parameterized dataset $\mathcal{D}_{\boldsymbol{\lambda}}$ as follows. Suppose the data consists of samples $(X, Y, G)$, where $G \in \mathcal{G}$ is the group label and $\mathcal{G}$ is the set of all groups present in the data. We denote by $\boldsymbol{\lambda}$ a $|\mathcal{G}|$-dimensional hyperparameter combination that lies in the $|\mathcal{G}|$-1 probability simplex. The vector $\boldsymbol{\lambda}$ consists of the probabilities of each group appearing in $\mathcal{D}_{\boldsymbol{\lambda}}$. To create $\mathcal{D}_{\boldsymbol{\lambda}}$, we first sample the group membership label according to $\boldsymbol{\lambda}$, and then uniformly sample an example from the chosen group, allowing for repetition in sampling. Consequently, $\boldsymbol{\lambda}$ is a $|\mathcal{G}|$-dimensional hyperparameter that controls the proportion of each group in $\mathcal{D}_{\boldsymbol{\lambda}}$. The dataset is considered evenly balanced across groups when all probabilities are equal to $1/|\mathcal{G}|$. It is equivalent to the original dataset when $\boldsymbol{\lambda}$ matches the prior probability over $\mathcal{G}$. We define two objective functions: (i) the average error $\ell_{\text{err}}(\boldsymbol{\lambda}) = \mathbb{E}\left[\mathbb{1}_{f_{\boldsymbol{\lambda}}(X)\neq Y}\right]$, and (ii) the worst error over all subgroups $\ell_{\text{worst-err}}(\boldsymbol{\lambda}) = \max_{g \in \mathcal{G}} \mathbb{E}\left[\mathbb{1}_{f_{\boldsymbol{\lambda}}(X)\neq Y}|G = g\right]$.

**Robust and Selective Classification.** We also examine the case of *selective* classification and robustness. The selective classifier can abstain from making a prediction when it is unsure (Geifman & El-Yaniv, 2017). Specifically, the model abstains when its confidence is lower than a threshold $\tau$, i.e. $f_{\boldsymbol{\lambda}}(x) < \tau$, where $f_{\boldsymbol{\lambda}}(x)$ denotes the probability of the predicted class. This approach can potentially enhance prediction performance by trading-off coverage, defined as the proportion of the population for which the classifier provides a prediction. In this case, we have a $|\mathcal{G}| + 1$-dimensional hyperparameter $\boldsymbol{\lambda}' = (\boldsymbol{\lambda}, \tau)$ and an additional objective function of the mis-coverage rate $\ell_{\text{mis-cover}}(\boldsymbol{\lambda}') = \mathbb{E}\left[\mathbb{1}_{f_{\boldsymbol{\lambda}}(x)<\tau}\right]$.

**VAE.** Variational Autoencoders (VAEs) (Kingma & Welling, 2013; Rezende et al., 2014) are generative models that leverage a variational approach to learn the latent variables underlying the data, and can generate new samples by sampling from the latent prior distribution. We focus on a $\beta$-VAE (Higgins et al., 2016), which balances the reconstruction error (distortion) and the Kullback Leibler (KL) divergence (rate):

$$R(f; \beta) = \mathbb{E}_{p_d(\mathbf{x})}\left[\mathbb{E}_{q_\phi(\mathbf{z}|\mathbf{x})}\left[-\log p_\theta(\mathbf{x}|\mathbf{z})\right]\right] + \beta \cdot \mathbb{E}_{p_d(\mathbf{x})}\left[D_{KL}(q_\phi(\mathbf{z}|\mathbf{x})||p(\mathbf{z}))\right], \tag{17}$$

where $\mathbf{z} \in \mathbb{R}^D$ is the latent embedding, $f$ consists of an encoder $q_\phi(\mathbf{z}|\mathbf{x})$ and a decoder $p_\theta(\mathbf{x}|\mathbf{z})$, parameterized by $\phi$ and $\theta$, respectively, and $p(\mathbf{z})$ is the latent prior distribution. Generally, models with low distortion

perform high-quality reconstruction but generate less realistic samples and vice versa. We define the hyperparameter $\boldsymbol{\lambda} = (\beta, D)$ consisting of the KL penalty strength $\beta$ and the latent dimension $D$. We specify two objectives $\ell_{\text{recon}}(f)$ and $\ell_{\text{KLD}}(f)$ defined by the left and right terms in (17), respectively.

**Transformer Pruning.** We adopt the multi-dimensional transformer pruning scheme proposed in (Laufer-Goldshtein et al., 2023), which involves three strategies for reducing computational complexity: (i) token pruning, removing unimportant tokens from the input sequence, (ii) layer early-exiting, computing part of the model's layers for easy examples, and (iii) head pruning, removing a portion of attention heads from the model architecture. We obtain $\boldsymbol{\lambda} = (\lambda_1, \lambda_2, \lambda_3)$ with the three thresholds controlling the pruning strength in each dimension, and consider two objectives: (i) the accuracy difference between the full model and the pruned model $\ell_{\text{diff-acc}}(\lambda) = \mathbb{E}\left[(\mathbb{1}_{f(X)=Y} - \mathbb{1}_{f_{\boldsymbol{\lambda}}(X)=Y})_+\right]$ and (ii) the respective cost ratio $\ell_{\text{cost}}(\lambda) = \mathbb{E}\left[\frac{C(f_{\boldsymbol{\lambda}}(X))}{C(f(X))}\right]$.

**Early Time Classification.** We adapt the early time classification scheme proposed by Ringel et al. (2024) for predicting the label of a given input data stream as quickly as possible. Specifically[3], we focus on employing LLMs for the task of reading comprehension, where the goal is to analyze a long document (given as context) and select an answer to a provided question. Let $\pi^t(X)$ denote a heuristic confidence measure of the prediction made based on the input $X$ received until time $t$ (e.g. the maximum predicted probability). We define the stopping-time for input $X$ as $\tau(X) = \{\min_t : \pi^t(X) \geq \lambda_t \text{ or } t = t_{\max}\}$. In this setting, the hyperparameter $\boldsymbol{\lambda} = (\lambda_1, \lambda_2, \ldots, \lambda_{t_{\max}})$ consists of the thresholds for all possible stopping times $t = 1, \ldots, t_{\max}$. We have two objectives: (i) the accuracy difference between the full-time prediction and the early-time prediction $\ell_{\text{diff-acc}}(\lambda) = \mathbb{E}\left[(\mathbb{1}_{f(X)=Y} - \mathbb{1}_{f_{\boldsymbol{\lambda}}(X)=Y})_+\right]$ and (ii) the normalized halt time $\ell_{\text{time}}(\lambda) = \mathbb{E}\left[\tau(X)/t_{\max}\right]$.

# 7 Experiments

We describe the experimental setup and present our main results. Further experimental details, as well as additional results are provided in Appendixes D and E, respectively. In the experiments we demonstrate the efficiency of the proposed method compared to baselines in two main ways: (i) For a fixed budget, we show that GuideBO achieves the lowest values for the free objective function compared to baselines; (ii) For a fixed level of the objective function, we show that GuideBO requires the smallest budget to achieve that level. Additional ablation studies are conducted to highlight the robustness of GuideBO.

## 7.1 Baselines

We define several baselines. We emphasize that in the second testing stage, both GuideBO and the baselines follow the same testing procedure that guarantees risk control. The baselines differ only in their optimization mechanisms during the first stage; thus, they can all be considered variants of Pareto Testing (Laufer-Goldshtein et al., 2023). We define two simple baselines and three multi-objective optimizers aimed at recovering the full Pareto front:

- UNIFORM - defines a uniform grid of configurations in the hyperparameter space.
- RANDOM - a uniform random sampling for $n = 1$, and Latin Hypercube Sampling (LHS) (McKay et al., 2000) for $n > 1$.
- HVI - uses the same acquisition function as in GuideBO, defined in Eq. (4). The key difference is that the reference point is defined in the standard way by the maximum possible loss values, rather than our focused reference point (12).
- EHVI (Emmerich et al., 2006) - similar to HVI but includes uncertainty in the hypervloume computation. Here too the reference point is defined by the maximum possible loss values.
- PAREGO (Knowles, 2006; Cristescu & Knowles, 2015) - uses random scalarization with Tchebycheff function to convert the multi-objective function into a single-objective, then employs EI as the acquisition function. We use the SMAC3 implementation (Lindauer et al., 2022).

---

[3]In addition, We provide results for the task of early time classification of structured time series data, as described in Appendix D.

Table 2: Tasks Details

| Task | n | $(\ell_1, \ldots, \ell_{\text{free}})$ | (best $\ell_1$, …, worst $\ell_{\text{free}}$) | (worst $\ell_1$, …, best $\ell_{\text{free}}$) | $N$ | $N_0$ |
|---|---|---|---|---|---|---|
| Fairness | 2 | (Err., DDP) | (0.154, 0.145) | (0.225, 0.01) | 10 | 5 |
| Robustness | 4 | (Avg. Error, Worst Err.) | (0.045, 0.62) | (0.089, 0.11) | 30 | 20 |
| Selective & Robust. | 5 | (Avg. Error, Mis-cover., Worst Err.) | (0.045, 0.0, 0.62) | (0.089, 0.0, 0.11) | 30 | 20 |
| VAE | 2 | (Recon. Err., KLD) | (0.001, 88) | (0.07, 0.001) | 10 | 5 |
| Pruning | 3 | (Acc. Difference, Rel. Cost) | (0.0, 1.0) | (0.8, 0.0) | 50 | 30 |
| Early-Time Class. | 10 | (Acc. Difference, Halt Time) | (0.0, 1.0) | (0.12, 0.1) | 50 | 30 |

## 7.2 Datasets

Here we describe the datasets used for each task. Table 1 summarizes the number of samples for each dataset according to the different splits (train/validation/calibration/test). We use the following datasets:

**Fairness.** We use the `Adult` (Dua et al., 2017) dataset, which consists of samples of individuals with 14 features as an input. The goal is to predict whether their annual income is above 50k$. Gender is considered as a sensitive attribute.

**Robustness + Robust and selective classification.** We use `CelebA` (Lin et al., 2019) and consider a binary prediction task of whether a person has a blond hair. The spurious correlation is associated with the gender at-

Table 1: Datasets Details

| Dataset | Train | Validation | Calibration | Test |
|---|---|---|---|---|
| Adult | 32,559 | 3,618 | 4,522 | 4,523 |
| CelebA | 162,770 | 19,867 | 9,981 | 9,981 |
| MNIST | 50,000 | 10,000 | 5,000 | 5,000 |
| AG News | 120,000 | 2,500 | 2,500 | 2,600 |
| Quality | - | 1,537 | 1,536 | 1,536 |

tribute, resulting in $|\mathcal{G}| = 4$ groups: (blond, female), (blond, male), (non-blond, female), (non-blond, male).
**VAE.** We use the `MNIST` dataset (LeCun, 1998), which consists of grayscale images of handwritten digits.
**Pruning.** We use `AG News` (Zhang et al., 2015) dataset where the task is to predict the category (out of four options) of news articles based on their content.
**Early-Time Classification.** We use the `QuALITY` dataset (Pang et al., 2022), which consists of triplets with a question, multiple choice answers, and a long context, along with the corresponding correct choice. The long context is partitioned into $t_{\max} = 10$ segments.

## 7.3 Evaluation

We emphasize again the purpose of each data split. The training data is used to learn the model parameters. The validation data is used for selecting candidate hyperparameter configurations, using either GuideBO (Algorithm 1) or the baseline procedures. The same data is also used for ordering the chosen configurations before testing. The calibration data is used for the FST procedure over the ordered set of chosen configurations. The final selected $\boldsymbol{\lambda}^*$ (14) determines the model setup. Lastly, the performance of the selected model is assessed on the test dataset.

Since the same testing procedure is applied across all methods, the chosen configuration is guaranteed to satisfy the specified constraints, as we verify empirically (see Fig. E.1). Therefore, our primary metrics for evaluating the efficiency of each method are: (i) its ability to minimize the free objective function within a given budget, and (ii) its capacity to reach certain levels of the free objective with the least budget.

We repeat the experiments with 5 random seeds over the optimization procedure. For each seed, we further generate 20 random splits of calibration and test subsets. Accordingly, we obtain $5 \times 20 = 100$ random trials, and report the mean and standard deviation across all trials. For each task, we choose the values of $\alpha$ according to the objective values obtained for the initially generated configurations. Table 2 lists the range values for each objective. We select values that lie within the range defined by these extreme points, ensuring they are not too close to either boundary. This is because values that are too small may not be statistically achievable, while excessively large values can be trivially satisfied, with tighter control not significantly improving the free objective. We set $\delta = 0.1$ and $\gamma = 0.01$.

## 7.4 Results

**Minimization of the free objective function for a given budget.** We examine the following scenarios: **Fairness** - error is controlled and DDP is minimized; **Robustness** - avg. er-

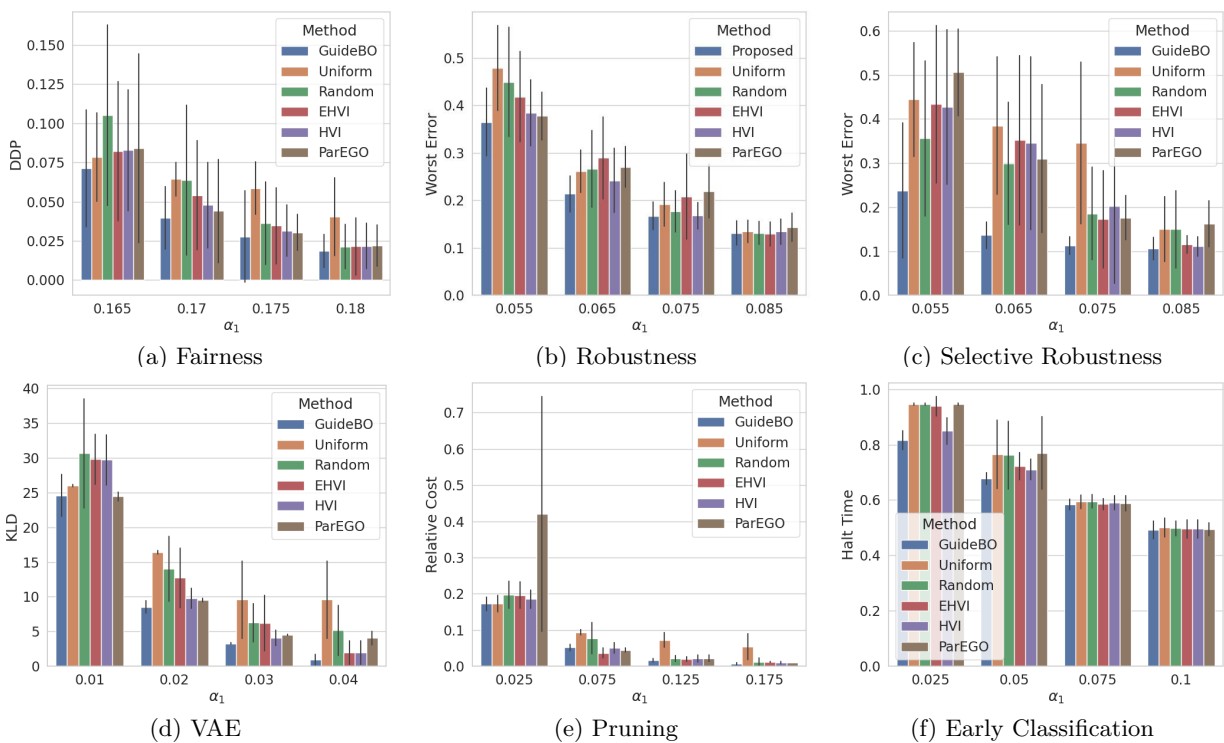

(a) Fairness        (b) Robustness        (c) Selective Robustness

(d) VAE        (e) Pruning        (f) Early Classification

Figure 4: The values of the free objective functions across tasks and different limits. The objectives are evaluated over $\mathcal{D}_{\text{test}}$ for the configuration that was chosen by each method. GuideBO consistently surpasses the baselines in nearly all cases. In contrast, the performance of the baselines exhibits significant variability.

ror is controlled and worst error is minimized; **Robustness and selective classification** - error and miscoverage are controlled while worst error is minimized; **VAE** - reconstruction error is controlled and KLD is minimized; **Pruning** - error difference is controlled and relative cost is minimized; **Early time classification** - error difference is controlled and relative cost is minimized.

Results are presented in Fig. 4 showing the values of the free objective function, evaluated over test data, across all tasks and $\alpha$ levels. To summarize the results, we rank the methods in each scenario, and report the counts of all rankings, as well as the average rank in Fig. 5. We observe that GuideBO consistently outperforms all baselines in nearly all cases. The multi-objective baselines outperform the simple baselines that distribute configurations across the entire space. Moreover, the baselines exhibit inconsistent performance, delivering satisfactory results for certain tasks or specific $\alpha$ values, but falling short in others. This inconsistency can be attributed to the arbitrary distribution of the configurations for the baselines. As a result, we sometimes randomly obtain configurations that are close to the testing limit (thus efficient), while at other times, the closest configuration is relatively far (thus inefficient). In contrast, GuideBO achieves a dense sampling of the relevant part of the Pareto front, leading to more precise and stable control across various conditions.

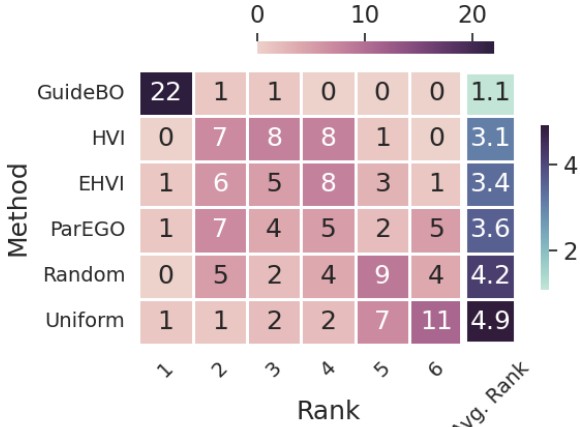

Figure 5: Rank count across settings and the average rank. GuideBO is ranked first in almost all cases, while the baselines have an inconsistent performance.

**Additional Results.** We first show that the constraints are satisfied in all cases by both GuideBO and the baselines in Fig. E.1. In addition, we show the objective values obtained by the different methods as a function of the budget (see Fig. E.2) and present the budget required to achieve specific objective levels (see Fig. E.3). The results show that GuideBO consistently outperforms the baselines. Furthermore, we explore the influence of varying the budget $N$ in comparison to a dense uniform grid (with 1000 points) in Fig. E.4. We show that $N = 100$ is sufficient to match (and sometimes outperform) the performance of the dense grid, highlighting the computational advantage of the proposed method. Furthermore, we conducted five additional experiments for the early-time classification task using various structured time-series datasets with large hyperparameter spaces ranging from 8 to 12 dimensions, demonstrating again that GuideBO is preferable over the baselines (see Table D.1 and Fig. E.5). We also examine the influence of the parameter $\gamma$ in Fig. E.6, showing that the method is generally insensitive to $\gamma$. Moreover, Fig. E.7 shows that using the proposed region is preferable over a single-sided upper bound at $\alpha$, implying that it is important to exclude inefficient configurations. Finally, we present examples of GuideBO's outcomes in Fig. E.8, highlighting its effectiveness in identifying relevant configurations within the defined region of interest, as opposed to recovering the entire front.

**Limitations and future work.** While our proposed method establishes an efficient mechanism for selecting risk-controlling model configurations and improves upon previous work, it has some limitations. Multi-fidelity optimization is a popular method for hyperparameter optimization, where resources are allocated efficiently (Li et al., 2018; 2020). In this approach, additional resources (e.g., more epochs) are allocated to promising configurations that performed well with fewer resources, while configurations that showed poorer performance are discarded. Our current method cannot be directly applied in this setting. Moreover, the calibrated selection procedure requires splitting the data between the validation set used for selecting the subset of promising configurations, and the calibration set used for their verification. Although this is commonly done in other conformal prediction and risk control methods (Angelopoulos et al., 2021; Bai et al., 2022; Ringel et al., 2024), in many practical settings there is only limited available data. Another issue is that there might be a distribution shift between the validation/calibration data and test data, and thus the selected configuration might not be risk-controlling with respect to shifted test data distributions (Gibbs & Candès, 2024; Zollo et al., 2024). These challenges should be explored in future work.

## 8   Conclusion

We introduce a versatile framework designed for reliable model selection. This framework is capable of meeting statistical risk limitations while simultaneously optimizing other conflicting metrics. We establish a confined region within the objective space that is a promising target for statistical testing. Our proposed method, referred to as GuideBO, is employed to pinpoint configurations that are Pareto optimal and lie in the specified region. We statistically validate the set of candidate configurations using multiple hypothesis testing to achieve verified control guarantees. The broad applicability and effectiveness of our approach are demonstrated for tuning different types of hyperparameters across various tasks and objectives, including high-accuracy, fairness, robustness, generation and reconstruction quality and cost and time considerations.

## Acknowledgments

We thank the anonymous reviewers for helpful discussions and feedback. B.L.G. was supported in part by Schmidt Sciences, LLC.

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

## A  Additional related work

**Gradient-Based MOO.**  When dealing with differentiable objective functions, gradient-based MOO algorithms can be utilized. The cornerstone of these methods is Multiple-Gradient Descent (MGD) (Sener & Koltun, 2018; Désidéri, 2012), which ensures that all objectives are decreased simultaneously, leading to convergence at a Pareto optimal point. Several extensions were proposed to enable convergence to a specific point on the front defined by a preference vector  (Lin et al., 2019; Mahapatra & Rajan, 2020), or learning the entire Pareto front, using a preference-conditioned model  (Navon et al., 2020; Lin et al., 2020; Chen & Kwok, 2022; Ruchte & Grabocka, 2021). However, this line of research focuses on differentiable objectives, optimizing the loss space used during training (e.g. cross entropy loss), which is typically different from the ultimate non-differentiable metrics used for evaluation (e.g. error rates). Furthermore, it focuses on recovering a single or multiple (possibly infinitely many) Pareto optimal points, without addressing the actual selection of model configuration under specific constraints, which is the problem we tackle in this paper.

**Incorporating user preferences in MOO.**  Another area of research has explored integrating decision maker preferences into multi-objective optimization (Branke, 2016; Wang et al., 2017; Xin et al., 2018). These methods are typically classified based on the stage in which they are applied in the optimization process: before, during, or after optimization (Branke & Deb, 2005). Preferences can be specified in various ways, such as using reference points representing an "ideal" solution (Deb & Sundar, 2006), imposing constraints (Fonseca & Fleming, 1998), setting maximal or minimal trade-offs (Branke et al., 2001), and using desirability functions that non-linearly scale the objectives to $[0, 1]$ (Wagner & Trautmann, 2010). Additionally, preferences can be learned by asking the decision maker to rank solutions or select the most preferred solution from a set (Fürnkranz & Hüllermeier, 2010). Another interactive preference learning approach involves incorporating explanations that help the decision maker in understanding the trade-offs between objectives (Misitano et al., 2022). In this paper, we focus on preferences provided by the user as hard limits on certain objectives, which require validation through statistical testing. Our BO procedure aims to extract a set of promising configurations that are designed to achieve tight control guarantees.

# B  Mathematical Details

## B.1  Derivation of the Region of Interest

Suppose the loss is bounded above by 1, then Hoeffding's inequality (Hoeffding, 1994) is given by:

$$\mathbb{P}\left(\hat{\ell}(\boldsymbol{\lambda}) - \ell(\boldsymbol{\lambda}) \leq -t\right) \leq e^{-2mt^2}, \tag{18}$$

and

$$\mathbb{P}\left(\hat{\ell}(\boldsymbol{\lambda}) - \ell(\boldsymbol{\lambda}) \geq t\right) \leq e^{-2mt^2}. \tag{19}$$

for $t > 0$. Taking $u = e^{-2mt^2}$, we have $t = \sqrt{\frac{\log(1/u)}{2m}}$, hence:

$$\mathbb{P}\left(\hat{\ell}(\boldsymbol{\lambda}) - \ell(\boldsymbol{\lambda}) \leq -\sqrt{\frac{\log\left(1/u\right)}{2m}}\right) \leq u, \tag{20}$$

and

$$\mathbb{P}\left(\hat{\ell}(\boldsymbol{\lambda}) - \ell(\boldsymbol{\lambda}) \geq \sqrt{\frac{\log\left(1/u\right)}{2m}}\right) \leq u. \tag{21}$$

This implies an upper confidence bound

$$\ell_{\mathrm{HF}}^{+}(\boldsymbol{\lambda}) = \hat{\ell}(\boldsymbol{\lambda}) + \sqrt{\frac{\log\left(1/u\right)}{2m}}, \tag{22}$$

and a lower confidence bound

$$\ell_{\mathrm{HF}}^{-}(\boldsymbol{\lambda}) = \hat{\ell}(\boldsymbol{\lambda}) - \sqrt{\frac{\log\left(1/u\right)}{2m}}. \tag{23}$$

In addition, we can use Hoeffding's inequality to derive a valid p-value under the null hypothesis $H_{\boldsymbol{\lambda}} : \ell(\boldsymbol{\lambda}) > \alpha$. By (20), we get:

$$\mathbb{P}\left(\hat{\ell}(\boldsymbol{\lambda}) - \alpha \leq -\sqrt{\frac{\log\left(1/u\right)}{2m}}\right) \leq \mathbb{P}\left(\hat{\ell}(\boldsymbol{\lambda}) - \ell(\boldsymbol{\lambda}) \leq -\sqrt{\frac{\log\left(1/u\right)}{2m}}\right) \leq u. \tag{24}$$

Rearranging the inequality inside the probability for $\hat{\ell}(\boldsymbol{\lambda}) < \alpha$, we obtain:

$$\begin{aligned}
\hat{\ell}(\boldsymbol{\lambda}) - \alpha &\leq -\sqrt{\frac{\log\left(1/u\right)}{2m}} \\
\alpha - \hat{\ell}(\boldsymbol{\lambda}) &\geq \sqrt{\frac{\log\left(1/u\right)}{2m}} \\
2m(\alpha - \hat{\ell}(\boldsymbol{\lambda}))^2 &\geq \log\left(1/u\right) \\
-2m(\alpha - \hat{\ell}(\boldsymbol{\lambda}))^2 &\leq \log\left(u\right) \\
e^{-2m(\alpha - \hat{\ell}(\boldsymbol{\lambda}))^2} &\leq u.
\end{aligned} \tag{25}$$

Thus for $\hat{\ell}(\boldsymbol{\lambda}) < \alpha$, we obtain:

$$\mathbb{P}\left(e^{-2m(\alpha - \hat{\ell}(\boldsymbol{\lambda}))^2_+} \leq u\right) \leq u, \tag{26}$$

where we inserted $(\cdot)_+ = \max(\cdot, 0)$ since we assume $\hat{\ell}(\boldsymbol{\lambda}) < \alpha$. Note that this statement also holds for $\hat{\ell}(\boldsymbol{\lambda}) \geq \alpha$, as we have $\mathbb{P}\left(1 \leq u\right) \leq u$, which is valid $\forall u \in [0,1]$. This implies that $p_{\boldsymbol{\lambda}}^{\mathrm{HF}} := e^{-2m\left(\alpha - \hat{\ell}(\boldsymbol{\lambda})\right)^2_+}$ is

super-uniform, hence is a valid p-value as required. Comparing $p_{\boldsymbol{\lambda}}^{\text{HF}}$ to $\delta$, yields the maximum empirical loss $\hat{\ell}(\boldsymbol{\lambda})$, evaluated over a calibration set of size $m$, which can pass the test with significance level $\delta$:

$$\alpha^{\text{max}} = \alpha - \sqrt{\frac{\log(1/\delta)}{2m}}. \tag{27}$$

This can be equivalently obtained from the upper bound defined in Eq. (22).

The region of interest in Eqs. (7) and (8) is obtained based on Eqs. (20) and (21) (where $u \to \gamma$ and $m \to k$):

$$
\begin{aligned}
&\mathbb{P}\left(\hat{\ell}_1^{\text{opt}}(\boldsymbol{\lambda}) \in R(\alpha, k, m, \delta, \gamma) \big| \ell_1(\boldsymbol{\lambda}) = \alpha^{\text{max}}\right) = \\
&\mathbb{P}\left(\ell_1(\boldsymbol{\lambda}) - \sqrt{\frac{\log(1/\gamma)}{2k}} < \hat{\ell}_1^{\text{opt}}(\boldsymbol{\lambda}) < \ell_1(\boldsymbol{\lambda}) + \sqrt{\frac{\log(1/\gamma)}{2k}}\right) = \\
&1 - \underbrace{\mathbb{P}\left(\hat{\ell}_1^{\text{opt}}(\boldsymbol{\lambda}) \leq \ell_1(\boldsymbol{\lambda}) - \sqrt{\frac{\log(1/\gamma)}{2k}}\right)}_{\text{Eqs. (20): } \leq \gamma} - \underbrace{\mathbb{P}\left(\hat{\ell}_1^{\text{opt}}(\boldsymbol{\lambda}) \geq \ell_1(\boldsymbol{\lambda}) + \sqrt{\frac{\log(1/\gamma)}{2k}}\right)}_{\text{Eqs. (21): } \leq \gamma} \geq 1 - 2\gamma.
\end{aligned}
\tag{28}
$$

Since the probability defined on the left-hand side in non-negative and bounded by 1, we have $0 < \gamma \leq 0.5$. In practice, we use much smaller values of $\gamma \ll 0.5$ to ensure a sufficiently wide region that encompasses all potential configurations around $\alpha^{\text{max}}$.

A tighter alternative to Hoeffding p-value was proposed in (Bates et al., 2021) based Hoeffding and Bentkus inequalities. The Hoeffding-Bentkus p-value is given by:

$$p_{\boldsymbol{\lambda}}^{\text{HB}} = \min\left(\exp\{-mh_1(\hat{\ell}(\boldsymbol{\lambda}) \wedge \alpha, \alpha)\}, e\mathbb{P}\left(\text{Binom}(m, \alpha) \leq \lceil m\hat{\ell}(\boldsymbol{\lambda})\rceil\right)\right), \tag{29}$$

where $h_1(a, b) = a\log(\frac{a}{b}) + (1 - a)\log(\frac{1-a}{1-b})$, $a \wedge b = \max(a, b)$ and $e$ denotes the natural exponent. For binary risk functions (e.g. error) we use the Binomial tail probability instead (without the $e$ factor):

$$p_{\boldsymbol{\lambda}}^{\text{Bin}} = \mathbb{P}(\text{Binom}(m, \alpha) \leq \lceil m\hat{\ell}(\boldsymbol{\lambda})\rceil). \tag{30}$$

Note that for a given $\delta$ we can numerically extract from Eqs. (29) or (30) the upper and lower bounds corresponding to a $1 - 2\gamma$ confidence interval, and use it to define the region of interest as in Eq. (8).

When the loss is unbounded, we can alternatively use a p-value defined by the central limit theorem, which is asymptotically valid. Assuming that the loss has a finite mean and variance, we define:

$$p_{\boldsymbol{\lambda}}^{\text{CLT}} = 1 - \boldsymbol{\Phi}\left(\frac{\alpha - \hat{\ell}(\boldsymbol{\lambda})}{\hat{\sigma}(\boldsymbol{\lambda})/\sqrt{m}}\right), \tag{31}$$

where $\hat{\sigma}(\boldsymbol{\lambda}) = \frac{1}{m-1}\sqrt{\sum_{j=k+1}^{m+k}(\ell(X_j, Y_j; \boldsymbol{\lambda}) - \hat{\ell}(\boldsymbol{\lambda}))^2}$ denotes the empirical standard deviation, and $\Phi$ is the normal cumulative distribution function. We obtain that $\limsup_{m\to\infty} \mathbb{P}\left(p_{\boldsymbol{\lambda}}^{\text{CLT}} \leq u\right) \leq u$.

## B.2 A valid p-value for multiple constraints

We prove that taking the maximum p-value across constraints is a valid p-value for the combined hypothesis.

**Lemma B.1.** *Let $p_{\boldsymbol{\lambda}, i}$ be a p-value for $H_{\boldsymbol{\lambda}, i} : \ell_i(\boldsymbol{\lambda}) > \alpha_i$, for each $i \in \{1, \ldots, c\}$. Define $p_{\boldsymbol{\lambda}} := \max_{1 \leq i \leq c} p_{\boldsymbol{\lambda}, i}$. Then, for all $\boldsymbol{\lambda}$ such that $H_{\boldsymbol{\lambda}} : \exists i$ where $\ell_i(\boldsymbol{\lambda}) > \alpha_i$ holds, we have:*

$$\mathbb{P}\left(p_{\boldsymbol{\lambda}} \leq u\right) \leq u, \tag{32}$$

*where $u \in [0, 1]$.*

*Proof.* Let $\mathcal{J} \subseteq \{1, \ldots, c\}$ be the set of all true null hypotheses (unsatisfied constraints) at $\boldsymbol{\lambda}$. We have:

$$\mathbb{P}\left(p_{\boldsymbol{\lambda}} \leq u\right) \leq \mathbb{P}\left(\max_{j \in \mathcal{J}} p_{\boldsymbol{\lambda}, j} \leq u\right) = \mathbb{P}\left(\bigcap_{j \in \mathcal{J}} p_{\boldsymbol{\lambda}, j} \leq u\right) \leq \max_{j \in \mathcal{J}} \mathbb{P}\left(p_{\boldsymbol{\lambda}, j} \leq u\right). \tag{33}$$

Since for each $j \in \mathcal{J}$, $\mathbb{P}\left(p_{\boldsymbol{\lambda}, j} \leq u\right) \leq u$, we have $\max_{j \in \mathcal{J}} \mathbb{P}\left(p_{\boldsymbol{\lambda}, j} \leq u\right) \leq u$, implying that $\mathbb{P}\left(p_{\boldsymbol{\lambda}} \leq u\right) \leq u$. $\square$

### B.3 Proof of Proposition 5.1

The proof is based on (Angelopoulos et al., 2021; Laufer-Goldshtein et al., 2023), which we repeat here for completeness.

*Proof.* Recall that $\mathcal{D}_{\mathrm{val}}$ and $\mathcal{D}_{\mathrm{cal}}$ are two disjoint, i.i.d. datasets. Therefore, $\mathcal{D}_{\mathrm{cal}}$ is i.i.d. w.r.t the returned configuration set optimized in Algorithm 1 over $\mathcal{D}_{\mathrm{val}}$.

We now prove that the testing procedure returns a set of valid configurations with FWER bounded by $\delta$. Let $H_{\boldsymbol{\lambda}'}$ be the first true null hypothesis in the sequence. Given that $p_{\boldsymbol{\lambda}'}$ is a super uniform p-value under $H_{\boldsymbol{\lambda}'}$, the probability of making a false discovery at $\boldsymbol{\lambda}'$ is bounded by $\delta$. This means that the event that $H_{\boldsymbol{\lambda}'}$ is rejected (false discovery) occurs with probability lower than $\delta$. However, if $H_{\boldsymbol{\lambda}'}$ fails to be rejected (no false discovery), then all other $H_{\boldsymbol{\lambda}}$ that follow in the sequence also fail to be rejected (regardless of if $H_{\boldsymbol{\lambda}}$ is true or not). Therefore, the probability of making any false discovery is bounded by $\delta$, which satisfies the FWER control requirement.

$\square$

### B.4 Hypervolume

An illustration of the hypervolume defined in Eq. (3) is given in Fig. B.1 for the 2-dimensional case. It can be seen that the hypervolume is equivalent to the volume of the union of the boxes created by the Pareto optimal points and the reference point.

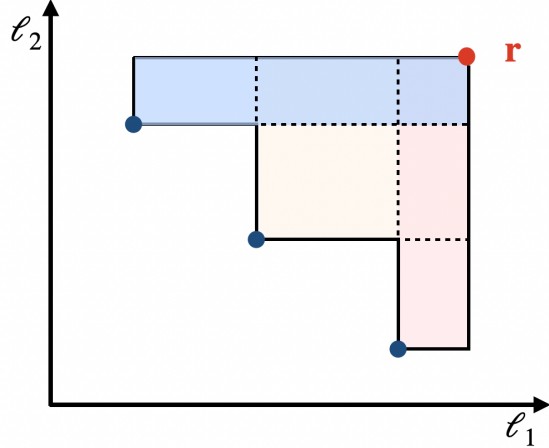

Figure B.1: An illustration of the hypervolume in the 2-dimensional case. The reference point is marked in red and three Pareto optimal points are marked in blue.

### B.5 Region of Interest

An illustration of the region of interest defined in Eq. (9) is given in Fig. B.2 for the 3-dimensional case (two constraints and a single free objective function). The volume is defined by the intersection of the regions defined by each constrained dimension.

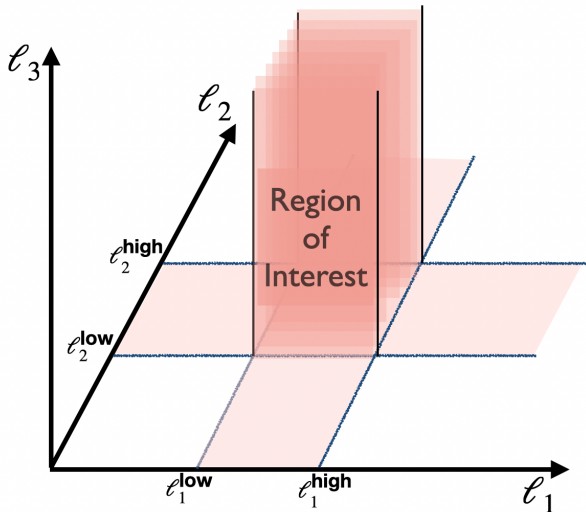

Figure B.2: An illustration of the region of interest in the 3-dimensional case.

## C Algorithms

Our overall proposed method is summarized in Algorithm C.1.

---

**Algorithm C.1** Configuration Selection

---

**Definitions:** $f$ is a configurable model set by an hyperparameter $\boldsymbol{\lambda}$. $\mathcal{D}_{\text{val}} = \{X_i, Y_i\}_{i=1}^k$ and $\mathcal{D}_{\text{cal}} = \{X_i, Y_i\}_{i=k+1}^{k+m}$ are two disjoint subsets of validation and calibration data, respectively. $\{\ell_1, \dots, \ell_c\}$ are constrained objective functions, and $\ell_{\text{free}}$ is a free objective. $\{\alpha_1, \dots, \alpha_c\}$ are user-specified bounds for the constrained objectives. $\Lambda$ is the configuration space. $\delta$ is the tolerance. $N$ is the optimization budget, and $N_0$ is the size of the intial pool of configurations. PARETOOPTIMALSET() returns Pareto optimal points.

1: **function** SELECT($\mathcal{D}_{\text{val}}, \mathcal{D}_{\text{cal}}, \Lambda, \{\alpha_1, \dots, \alpha_c\}, \delta, N$)
2:     Compute $\ell_i^{\text{low}}, \ell_i^{\text{high}}$ for $i \in \{1, \dots, c\}$ based on (8) and (9)        ▷ Determine the region of interest.
3:     $\mathcal{C}_0, \mathcal{L}_0 \leftarrow$ Randomly sample an initial pool of configurations of size $N_0$        ▷ Generate an initial pool.
4:     $\mathcal{C}^{\text{BO}} \leftarrow$ BO($\mathcal{D}_{\text{val}}, \ell, \mathcal{C}_0, \mathcal{L}_o, \{\ell_1^{\text{low}}, \dots, \ell_c^{\text{low}}\}, \{\ell_1^{\text{high}}, \dots, \ell_j^{\text{high}}\}, N$)        ▷ BO via Algorithm 1.
5:     $\mathcal{C}^{\text{P}} \leftarrow$ PARETOOPTIMALSET($\mathcal{C}^{\text{BO}}$)        ▷ Filter Pareto optimal points according to Eq. (2).
6:     Compute $p_{\boldsymbol{\lambda}}^{\text{val}}$ over $\mathcal{D}_{\text{val}}$ for all $\boldsymbol{\lambda} \in \mathcal{C}^{\text{P}}$        ▷ Compute approximated p-values.
7:     $\mathcal{C}^{\text{o}} \leftarrow$ Order configurations according to increasing $p_{\boldsymbol{\lambda}}^{\text{val}}$        ▷ Order configurations.
8:     Compute $p_{\boldsymbol{\lambda}}^{\text{cal}}$ over $\mathcal{D}_{\text{cal}}$ for all $\boldsymbol{\lambda} \in \mathcal{C}^{\text{o}}$        ▷ Compute p-values.
9:     Apply FST: $\mathcal{C}^{\text{valid}} = \{\boldsymbol{\lambda}^{(j)} : j < J\}, \ J = \min_j\{j : p_{\boldsymbol{\lambda}}^{\text{cal}} \geq \delta\}$        ▷ Apply FST.
10:     $\boldsymbol{\lambda}^* = \arg\min_{\boldsymbol{\lambda} \in \mathcal{C}^{\text{valid}}} \ell_{\text{free}}(\boldsymbol{\lambda})$        ▷ select the best-performing configuration.
11:     **return** $\boldsymbol{\lambda}^*$

---

## D Implementation and dataset details

We provide here further details on the datasets, application specifications, model architectures, training procedures, and examined scenarios.

**Initialization.** For GuideBO, HVI and EHVI we randomly sample an initial set of size $N_0$, and perform $N - N_0$ iterations of BO. We use a uniform grid for $n = 1$ and LHS for $n > 1$. The values of $N$ and $N_0$ for each task are provided in Table 2. For PAREGO we use the default initialization defined by the SMAC3 implementation.

**Fairness.** For computing $\widehat{\text{DDP}}(f)$, the indicator $\mathbb{1}_{f(x)>0}$ in Eq. (15) is relaxed using $\tanh(c \cdot \max(0, f(\mathbf{x})))$ with $c = 3$ (Padh et al., 2021). In addition, we define a linear interpolation in the input space: $x_t = t \cdot x_1 + (1 - t) \cdot x_{-1}$, for $t \in [0, 1]$ where $x_1$ and $x_{-1}$ represent samples with attributes $a = 1$ and $a = -1$,

respectively. The mixup regularization is defined by:

$$\widehat{\text{MixUP}}(f) = \mathbb{E}_t\left[|\mathbb{E}_X\left[\langle\nabla_x f(x_t), x_{-1} - x_1\rangle\right]|\right], \tag{34}$$

regularizing the expected inner product between the Jacobian on mixup samples and the difference $x_{-1} - x_1$ (Chuang & Mroueh, 2020). Our model is a 3-layer feed-forward neural network with hidden dimensions $[60, 25]$. We train all models using Adam optimizer with learning rate $1e-3$ for 50 epochs and batch size 256.

**Robustness.** We use a ResNet-50 model pretrained on ImageNet. Following (Izmailov et al., 2022), we train the models for 50 epochs with SGD with a constant learning rate of $1e-3$, momentum decay of 0.9, batch size 32 and weight decay of $1e-4$. We use random crops and horizontal flips as data augmentation. We use half of the CelebA validation data to train the last layer, and the other half for BO.

**VAE.** We use the implementation provided by (Chadebec et al., 2022) of a ResNet-based encoder and decoder, trained using AdamW optimizer with $\beta_1 = 0.91, \beta_2 = 0.99$, and weight decay 0.05. We set the learning to $1e-4$ and the batch size to 64. The training process consisted of 10 epochs. We use binary-cross entropy reconstruction loss for training the model, and the mean squared error normalized by the total number of pixels (728) as the reconstruction objective function for hyperparameter tuning.

**Pruning.** We use a BERT-base model (Devlin et al., 2018) with 12 layers and 12 heads per layer. We follow the recipe in (Laufer-Goldshtein et al., 2023) and attach a prediction head and a token importance predictor per layer. The core model is first finetuned on the task. We compute the attention head importance scores based on 5K held-out samples out of the training data. We freeze the backbone model and train the early-exit classifiers and the token importance predictors on the training data (115K samples).

Each prediction head is a 2-layer feed-forward neural network with 32 dimensional hidden states, and ReLU activation. The input is the hidden representation of the `[CLS]` token concatenated with the hidden representation of all previous layers, following (Wołczyk et al., 2021).

Similarly, each token importance predictor is a 2-layer feed-forward neural network with 32 dimensional hidden states, and ReLU activation. The input is the hidden representation of each token in the current layer and all previous layers (Wołczyk et al., 2021).

**Early-Time Classification.** We adapt the setup described in (Ringel et al., 2024), and use the processed model outcomes that appear in their implementation[4]. The context of each question is divided into sentences, which are grouped into $t_{\max} = 10$ sets. The input sequence until time $t$ is provided as a prompt that includes the context sentences up to timestep $t$, along with the question and its four options, labeled 'A', 'B', 'C', and 'D'. The prompt concludes with "`The answer is:\n\n`". The prompt is processed by the Vicuna-13B model.

We provide additional results for structured time series data, following (Ringel et al., 2024). The details of all datasets are summarized in Table D.1. Each dataset is partitioned into four distinct parts: 80% for training and the remaining 20% are equally divided to validation, calibration and test subsets. For each dataset we define an exist point every fixed number of timesteps (hop-size) and obtain a hyperparameter dimension $n$ ranging from 8 to 12. A standard LSTM is used for feature extraction with one recurrent layer with a hidden size of 32, except for `WalkingSittingStanding` where the model consists of 2 recurrent layers, each with a hidden size of 256. The output of the last recurrent layer is followed by two fully connected classification heads, one for classifying the label and the other for estimating the classification confidence. The loss consists of two terms: cross-entropy loss for the label, and binary cross entropy loss (weighted by 0.2) for whether the classifier is correct or not. The models are trained with Adam optimizer, with a learning rate of 0.001, and a batch size of 64.

# E   Additional Results

In this section, we describe additional experiments and results.

---

[4]https://github.com/liranringel/etc

Table D.1: Summary of structured time-series datasets.

| Dataset | # Features | # Classes | # Samples | # Timesteps | Hop Size | n |
|---|---|---|---|---|---|---|
| Tiselac | 10 | 9 | 99687 | 23 | 2 | 12 |
| ElectricDevices | 1 | 7 | 16637 | 96 | 8 | 12 |
| PenDigits | 2 | 10 | 10992 | 8 | 8 | 8 |
| Crop | 1 | 24 | 24000 | 46 | 4 | 12 |
| WalkingSittingStanding | 3 | 6 | 10299 | 206 | 20 | 11 |

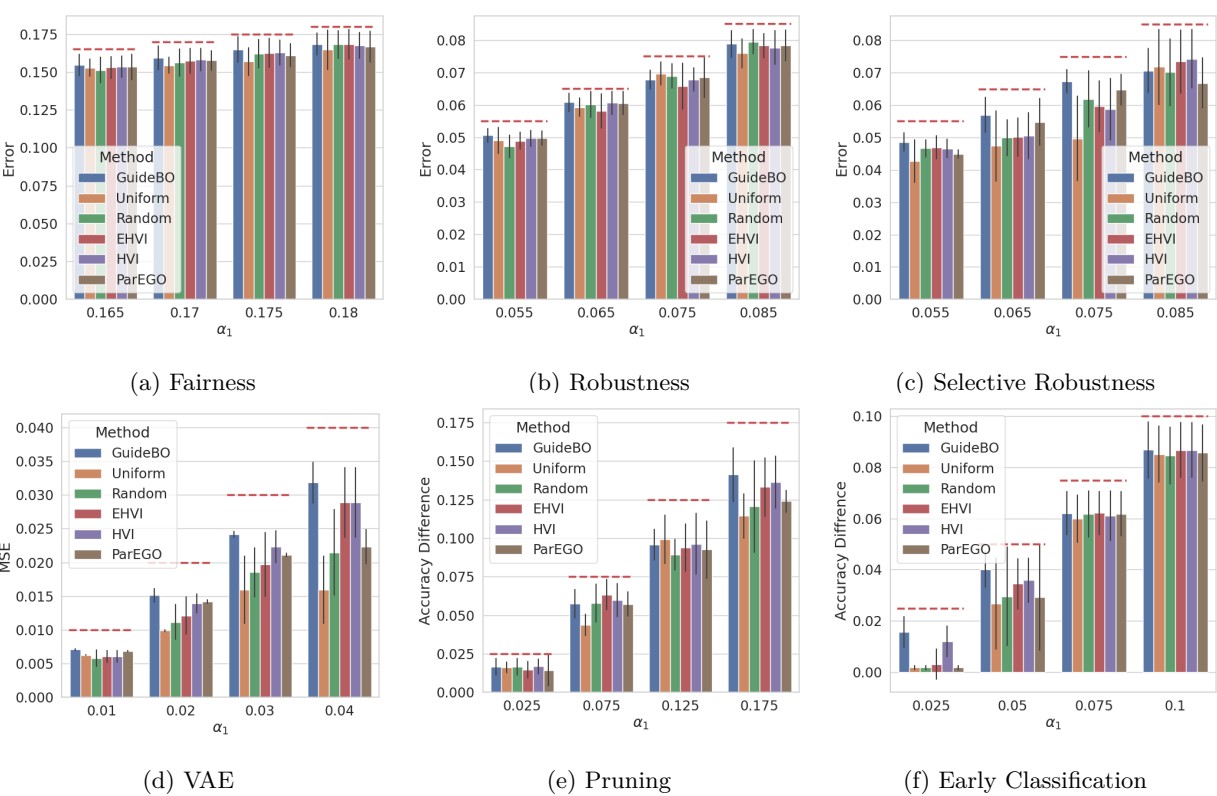

Figure E.1: The values of the constrained objective functions across tasks and different limits (marked by dashed red lines). The objectives are evaluated over $\mathcal{D}_{\text{test}}$ for the configuration that was chosen by each method. All method satisfy the limits due to the testing procedure.

**Satisfying Constraints and Tighter Control.** We show the values of the constrained objective functions in Fig. E.1, where the red dashed lines depict the limit. We see that the constraints are satisfied by all methods as expected, since in any case the configurations are validated through the testing procedure. Notably, GuideBO obtains tighter control compared to baselines in nearly all cases. This is consistent with our earlier finding that GuideBO better minimizes the free objective function.

**Varying Optimization Budget.** In this experiment we fixed the constraint ($\alpha$ limit) for each task and varied the optimization budget (number of iterations/function evaluations). We show the improvement in the free objective function as a function of the optimization budget across tasks on Fig. E.2. It can be seen that GuideBO delivers consistently strong performance in all scenarios, while the other baselines have mixed results, excelling in some cases and underperforming in others. In addition, for each task we define three levels of the free objective: between 90% of the maximum value and 110% of the minimum value obtained by the different baselines, and order these levels from high to low value. We compare the budget that is required for each baseline to reach a certain level in all tasks and for all defined levels. Results are shown in

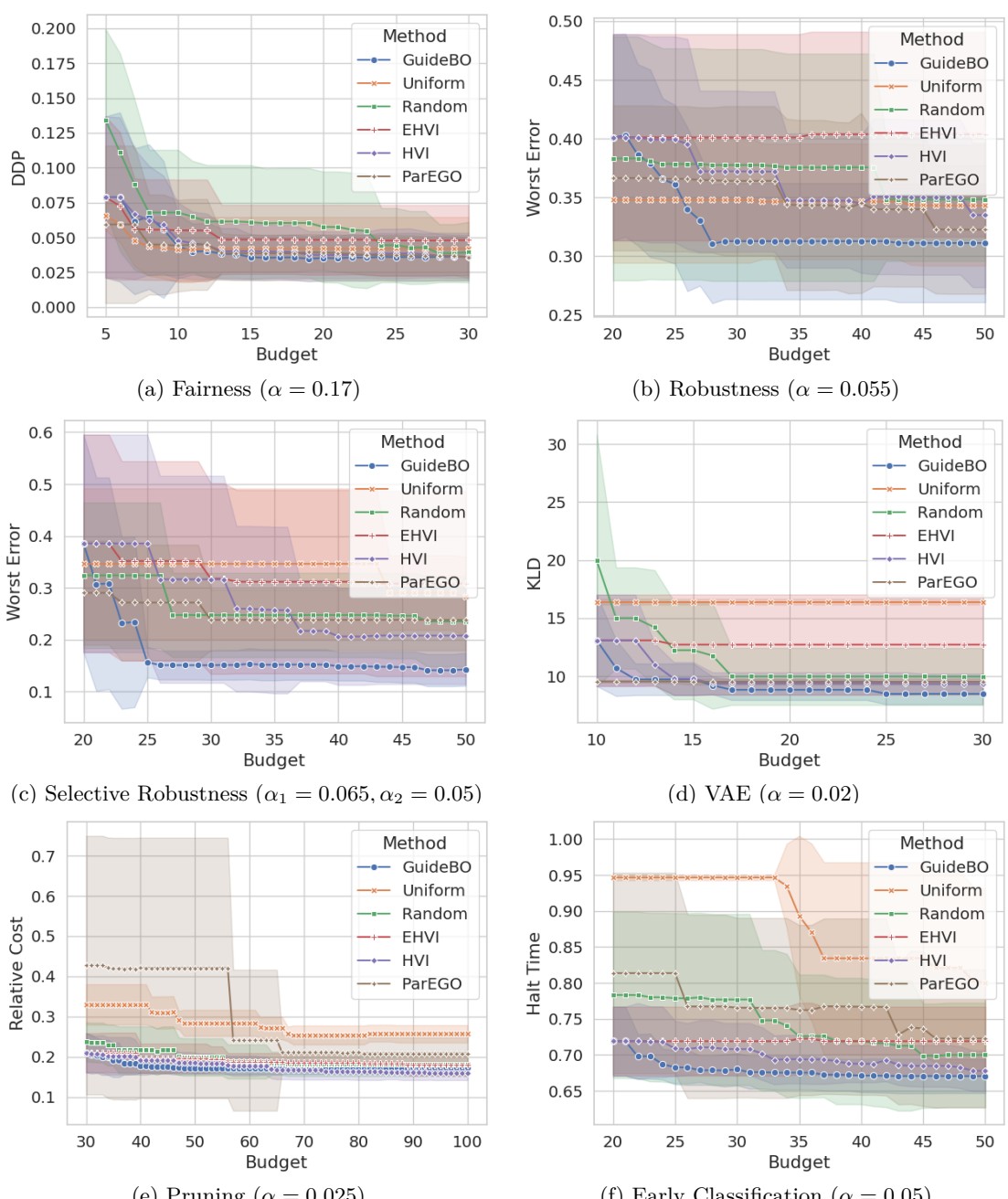

Figure E.2: The value of the free objective as a function of the budget across different tasks. In each case the $\alpha$ limit is fixed. GuideBO consistently performs well across all cases, whereas the other baselines show inconsistent performance, achieving better results in some instances and worse in others.

Fig. E.3. If a method cannot reach a certain level within the total maximum budget, no bar is displayed for that method at that level. We observe that in almost all cases (15 out of 18 scenarios), GuideBo requires the least budget compared to the baselines, and in many instances, it can reach levels that are unattainable by the other baselines. These results indicate that GuideBo requires fewer iterations to achieve the same or better results compared to the baselines.

**Comparing to dense grid.** We examine the effect of varying the optimization budget $N$. We show results for the pruning task with $N \in \{20, 50, 100\}$. In addition, we compare to a dense grid with uniform sampling

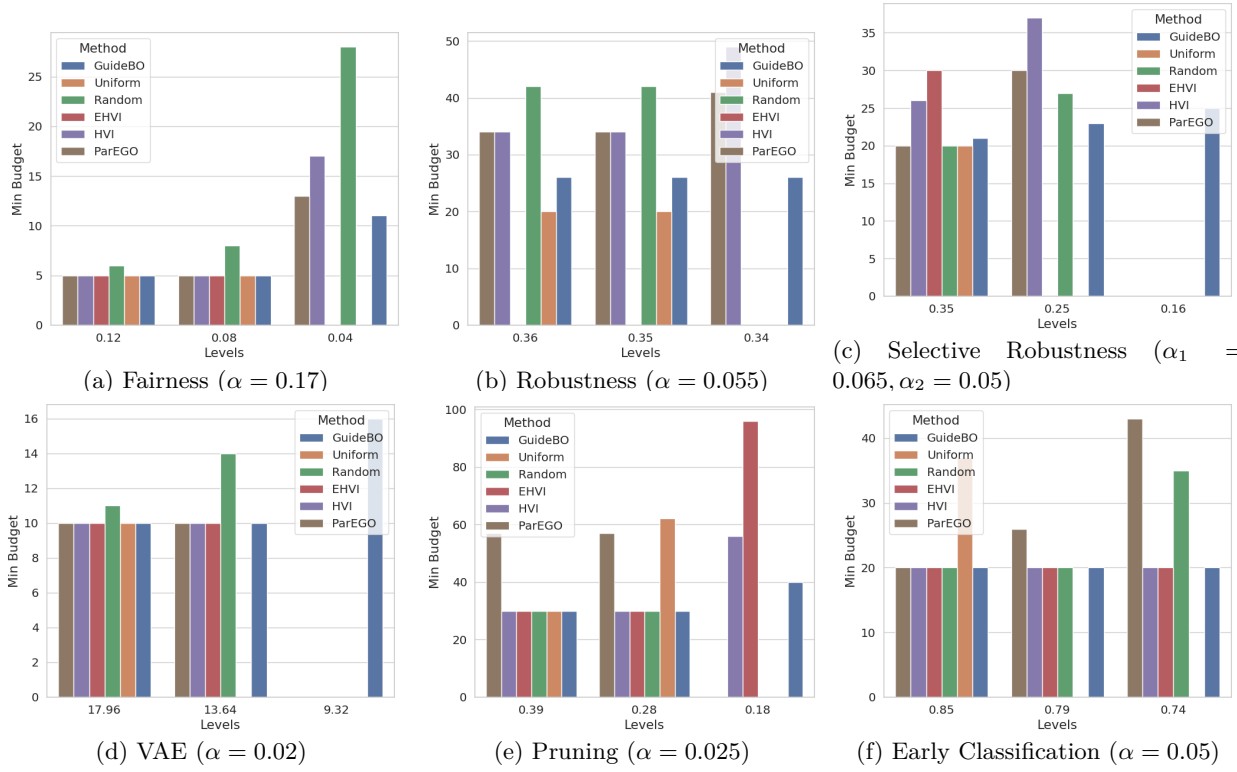

Figure E.3: The budget that is required to achieve several different levels of the free objective function across different tasks. In nearly all cases (15 out of 18 scenarios), we find that GuideBO requires the smallest budget compared to the baselines. Moreover, it often achieves levels that the other baselines cannot reach.

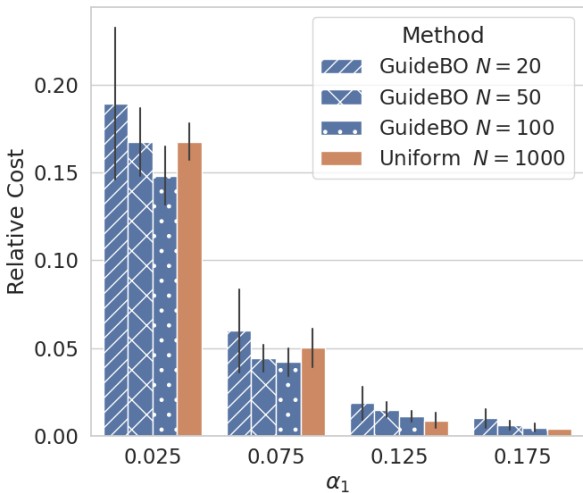

Figure E.4: Results of the proposed method over AG News (pruning task) for different number of evaluations, and with a grid of uniform thresholds. Accuracy reduction is controlled and cost is minimized.

of all 3 hyperparmeters with a total of $N = 1000$ configurations. We see on Fig. E.4 that the relative cost gradually improves with the increase in $N$. It reaches (and in some cases outperform) the dense grid baseline with $N = 100$ (that is 10% decrease in budget). This indicates that using our proposed method we can significantly decrease the required budget without scarifying performance.

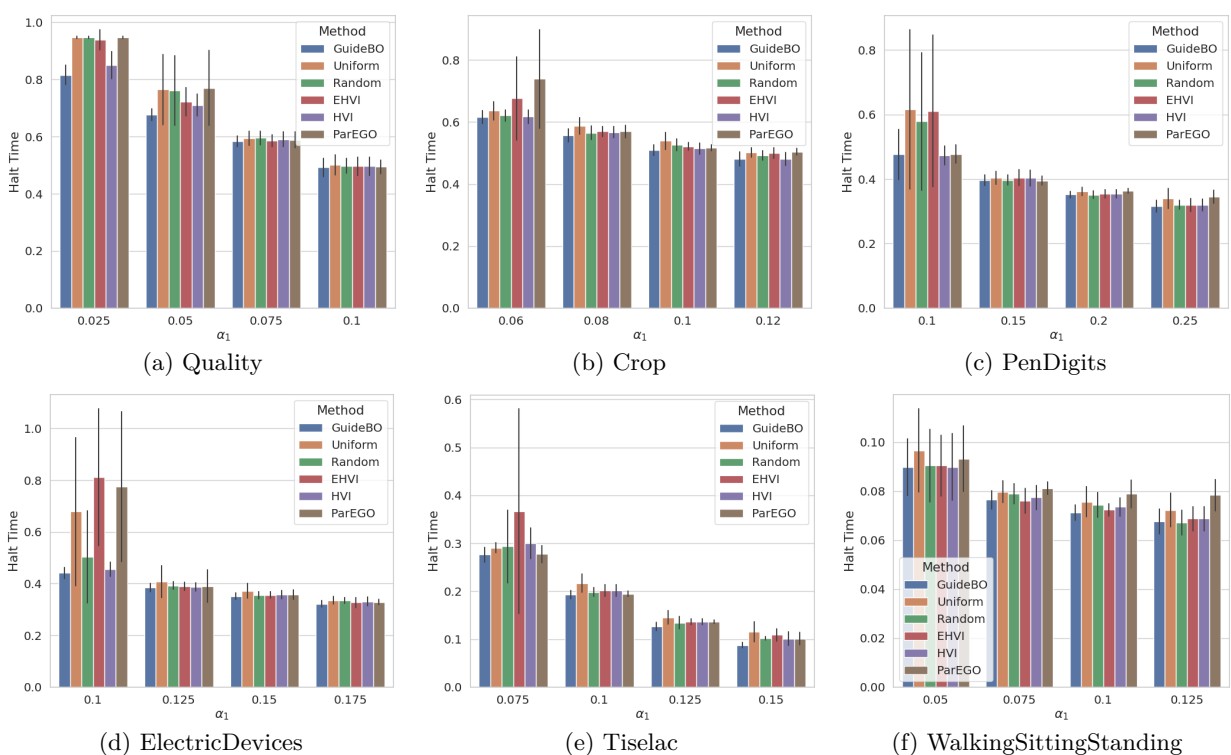

Figure E.5: The values of the free objective functions across datasets and different limits for the task of early time classification. GuideBO is ranked first in 20 out of 24 cases.

**Additional experiments for early-time classification task with different datasets.** We conducted additional experiments for the task of early time classification with different datasets. Following (Ringel et al., 2024), we show results on five additional structured time series datasets that are publicly available via the aeon toolkit[5]: Tiselac (Ienco & Gaetano, 2007), ElectricDevices (Chen et al., 2015), PenDigits (Alpaydin & Kaynak, 1998), Crop (Tan et al., 2017), and WalkingSittingStanding (Anguita et al., 2013). An LSTM model serves as the base sequential classifier for all datasets. Results of all experiments on early time classification (including Quality dataset) are presented on Fig. E.5. The average rank for each method across datasets and $\alpha$ levels is: GuideBO: 1.2, HVI: 2.8, Random: 3.2, EHVI: 3.4, ParEGO: 4.1, Uniform: 5.2. We conclude that, similar to our previous results, GuideBO outperforms the baselines in nearly all cases, ranking first in 20 out of 24 scenarios.

**Influence of $\gamma$.** We examine the influence of $\gamma$, which determines the boundaries of the region of interest. Figure E.6 shows the scores obtained for different values of $\gamma$. We observe that in most cases there is no noticeable difference in the performance with respect to $\gamma$. However, it appears that moderate values, neither too large nor too small, are preferable.

**Ablation study - one-sided upper bound.** We compare the proposed method to the case that the BO search is constrained by a one-sided bound at the upper limit defined by $\alpha$. This means that the reference point is set to $r_i = \alpha_i$ for $i \in 1, \ldots, c$, while $r_{c+1}$ is set according to the maximum value of $\ell_{\text{free}}$, as in the standard full Pareto front approach. Figure E.7 shows the values of the free objective across tasks. We see that in most cases performing the search in the defined region of interest is preferable to a single-sided bound. This shows the benefit of removing low risk, inefficient configurations from the search space (the green section in Fig. 2).

**Demonstration of BO Selection.** We show the outcomes of the BO procedure across several tasks in Fig. E.8. We compare the proposed method to HVI that recovers the entire Pareto front. The reference point defined in (12) is marked by a green square, and the boundaries of the region of interest are depicted

---

[5]https://www.aeon-toolkit.org/en/stable/

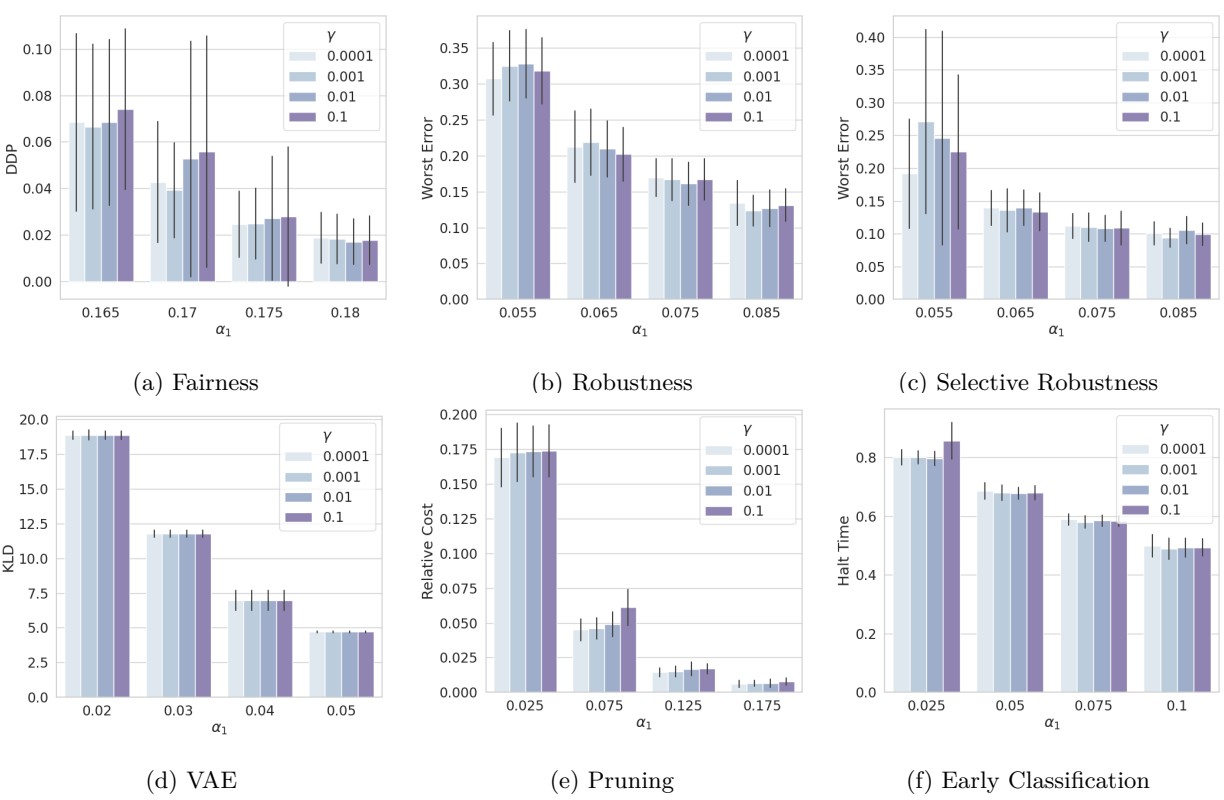

Figure E.6: Influence of $\gamma$. Showing the scores of the free objective for different values of $\gamma$, which controls the width of the region of interest, defined in Eq. (8).

by dashed lines. The blue points correspond to the configurations in the initial pool $\mathcal{C}_0$, while the red points correspond to the configurations selected by the BO procedure. We see that the specified region is significantly smaller compared to the entire front. Moreover, we observe that by GuideBO we obtain a dense set of configurations in the region of interest as desired. In contrast, for HVI we obtain samples all over the front. In addition, the distribution of points is not always evenly spread along the front, so that certain part of the front are denser than other. This explains why HVI (and similarly the other baselines) is inferior compared to GuideBO since for certain $\alpha$ values the distribution of the selected configurations is sparse near the limiting value.

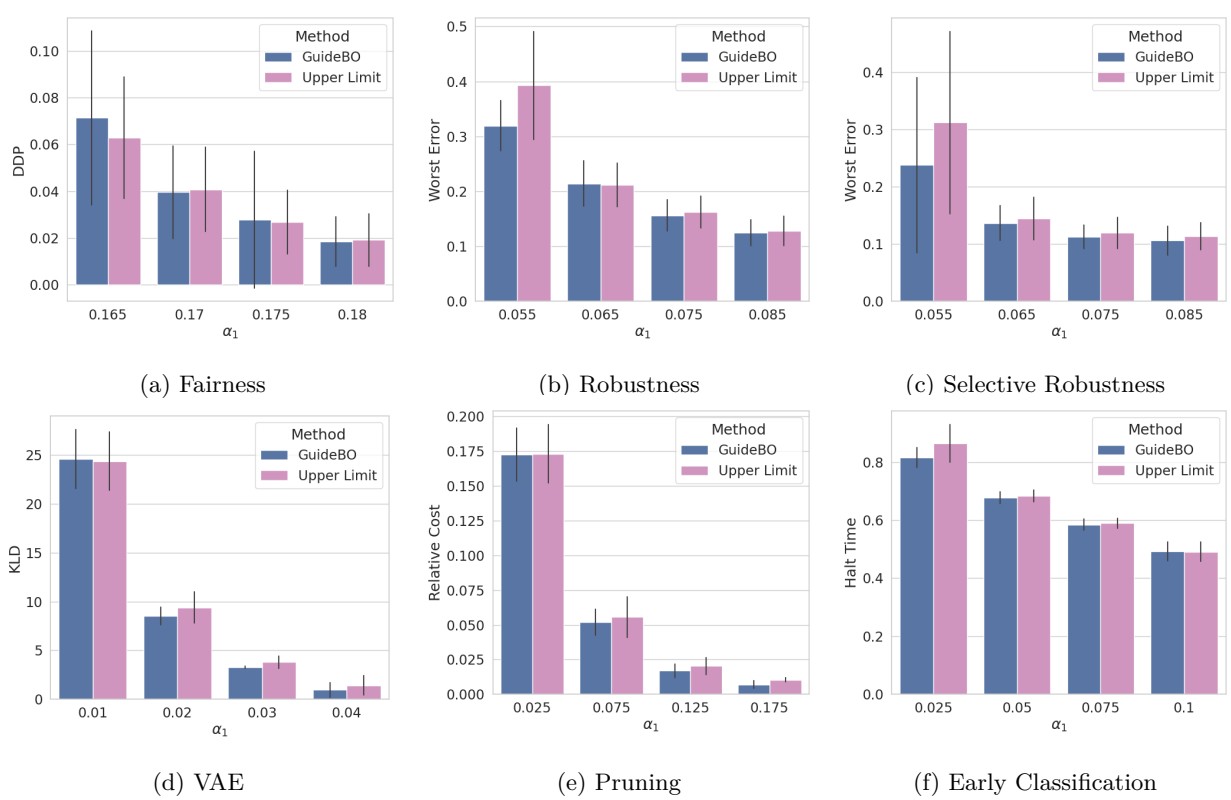

Figure E.7: Ablation study - comparing the proposed method with two-sided region to a one-sided upper bound, defined by the limit $\alpha$. The plots present the scores obtained for the free objective.

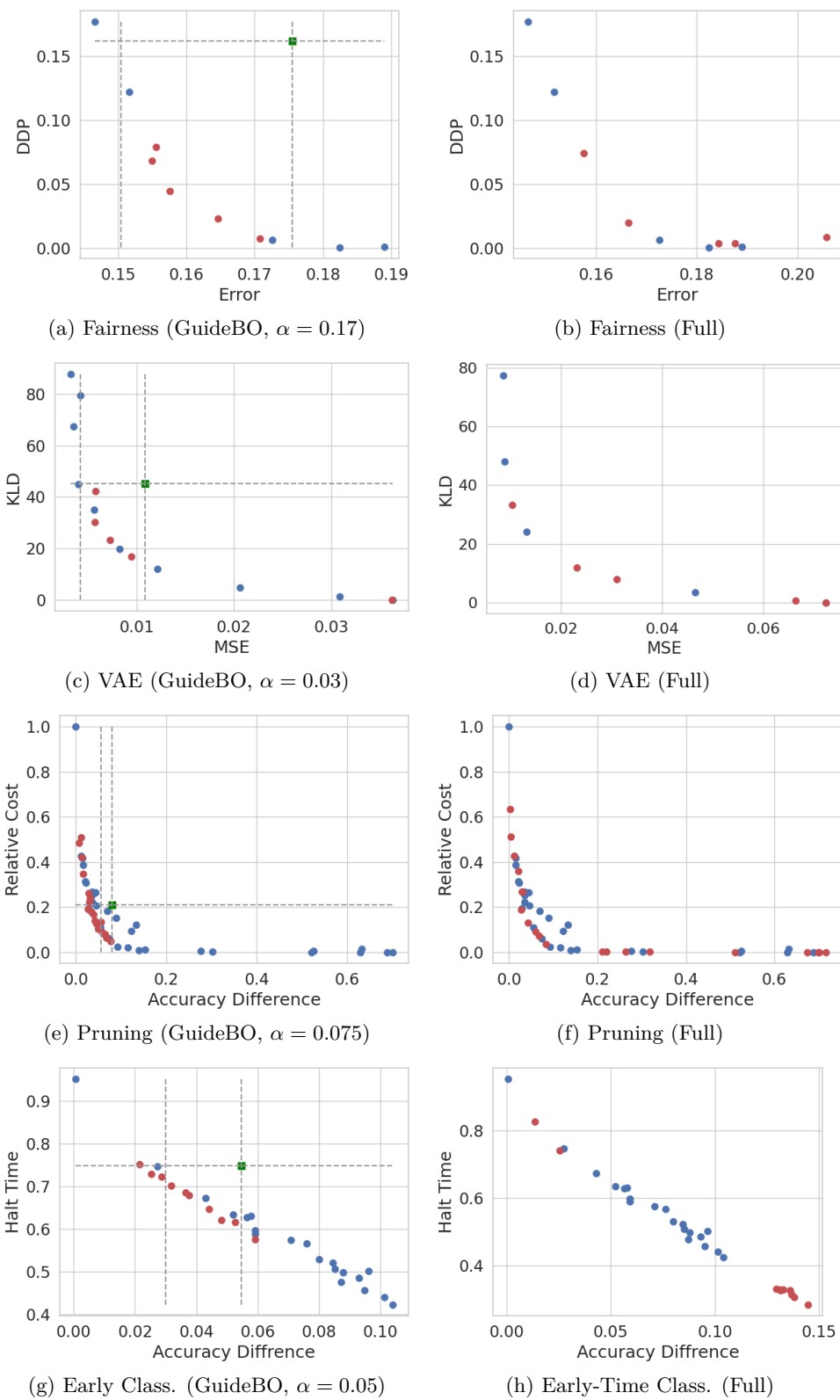

Figure E.8: Demonstration of the selection outcomes of the BO procedure, comparing the proposed method (left) to full recovery of the Pareto front by HVI (right): the green square is the defined reference point, the blue points correspond to the initial set of configurations, and the red points correspond to selected configurations. Dashed lines enclose the region of interest.

