# OpenReview forum: "Risk-Controlling Model Selection via Guided Bayesian Optimization"
_TMLR — Accepted by TMLR_

### Review · Reviewer_x3Gv · 2024-07-12

**Summary Of Contributions:**

The paper focuses on the problem of optimally tuning hyperparameters in machine learning by combining a Bayes-Optimal strategy aiming at identifying a subset of Pareto optimal model configurations related to a specific *region of interest*, aptly identified according to user-specified tolerances, subsequently selected according to a statistical hypothesis testing. The strategy is tested in a number of numerical experiments.

**Audience:**

Yes

**Broader Impact Concerns:**

There are no concerns about the ethical implications of the paper.

**Claims And Evidence:**

Yes

**Requested Changes:**

- The analysis relies on the adoption of a Gaussian process as a surrogate model. Do the authors expect this choice to be important for the efficacy of the algorithm? (e.g., is it possible that the algorithm might benefit from fat-tailed processes depending on the functional form of $\ell$?)
- On page 6, the authors assume that efficiency and validity are sort-of completely anti-correlated, and this justifies the assumption that the loss has to take values around $\alpha^{\rm max}$. This is not very clear to me: could the author clarify this step?
- Is the acquisition function optimization step possibly going to suffer non-convexity issues?
- From what I understand, it is assumed everywhere that $\boldsymbol\lambda\in\mathbb R^d$ with $d\ll m,k$. Is this correct?
- In the *Classification robustness* experimental setup, do the authors create $\mathcal D_{\boldsymbol\lambda}$ by allowing repetion in the sampling? In this case, it should be specified.

Minor aspects:
- The notation in Eq. 4 is not very clear (in particular, what $\boldsymbol\ell(\boldsymbol\lambda)\cup\hat{\mathcal P}$ means is not specified).
- On page 5, there is confusion in the notation regarding $\kappa$ and $k$.
- Right before Eq. 7, $\hat\ell_1^{\rm opt}$ does not seem to be defined.
- In line 1 of the Algorithm on page 8, do the authors mean $\ell_c^{\rm high}$?

**Strengths And Weaknesses:**

*Strength* — I found the paper quite clear and the proposed strategy, which combines a Bayesian strategy with a hypothesis testing step, seems to me an interesting combination and progress with respect to previous contributions. One of the key aspects of the method is the identification of a suitable feasible subspace of hyperparameters that allows the resizing of the space to be explored to search for Pareto configurations.

*Weaknesses* — The proposed strategy might result in a ```null``` set of hyperparameters, but this limitation might be avoided by choosing different input tolerances. As anticipated, the paper is globally well written; however, the portion on the numerical exploration of the strategy is more confusing as various experiments are presented and discussed in a quite compressed space.

---

> ### Author Response · Authors · 2024-08-01
> **Response**
>
> Thank you for your helpful and constructive review.
>
>
> > On confusion in the details of the experimental section.
>
>
> We added general clarifications at the beginning of the experimental section as well as rephrased cartian confusing descriptions.
>
> > On the adoption of a Gaussian process as a surrogate model and the benefits of using fat-tailed processes.
>
> In Bayesian optimization, Gaussian Processes are the most commonly used surrogate models due to their flexibility and tractability. However, there are several other surrogate models with more fat-tailed distributions that can be advantageous in certain situations. For example, Student-t processes are a generalization of Gaussian processes with heavier tails than Gaussian distributions [1]. Another possibility is using random forests and other ensemble tree methods that can provide robust predictions with fat-tailed characteristics [2]. Our framework is general and agnostic to specific optimization choices, allowing for the potential use of other surrogate models. We believe that the selection of these models is application-specific, and it would be an interesting direction for future research to explore how different models may impact risk control considerations.
>
>
> [1] Shah, Amar, Andrew Wilson, and Zoubin Ghahramani. "Student-t processes as alternatives to Gaussian processes." Artificial intelligence and statistics. PMLR, 2014.
>
> [2] Lindauer, Marius, et al. "SMAC3: A versatile Bayesian optimization package for hyperparameter optimization." Journal of Machine Learning Research, 2022.
>
> > On the assumption that efficiency and validity are completely anti-correlated.
>
> Indeed, in the paper we assume that the objectives are conflicting, so we focus on configurations near the limiting value to optimize efficiency with respect to the free objective. If the objectives are not conflicting, the situation can be treated as a constrained optimization problem. In such cases, the solution can be statistically tested and verified without concern for failing the test (since the solution minimizing the free objective will be away from the limit on the constrained objective). However, when objectives do conflict, the optimal solution will satisfy the constraints tightly. Thus, it is important to identify a set of configurations around the optimal point to increase the likelihood of finding a valid solution. We explore several tasks with conflicting objectives, such as accuracy versus fairness, accuracy versus robustness, accuracy versus computational cost, and reconstruction quality versus disentanglement of latent factors in variational autoencoders.
>
> > Is the acquisition function optimization step possibly going to suffer non-convexity issues?
>
> The Pareto front in a multi-objective optimization problem can be non-convex. This means that simple linear combinations of the objectives, a common technique in scalarization methods, might not effectively capture the trade-offs between objectives. However, HVI-based methods do not rely on scalarization. They directly consider the geometry of the Pareto front and aim to improve the hypervolume, making them inherently more robust to non-convex Pareto fronts. While it has been claimed that the HVI measure prefers convex regions over non-convex ones [1], it was shown in [2] that HVI-based methods also perform well in non-convex regions.
>
> [1] Zitzler, Eckart, and Lothar Thiele. "Multiobjective optimization using evolutionary algorithms—a comparative case study." International conference on parallel problem solving from nature. Berlin, Heidelberg: Springer Berlin Heidelberg, 1998.‏
>
> [2] Emmerich, Michael, Nicola Beume, and Boris Naujoks. "An EMO algorithm using the hypervolume measure as selection criterion." International Conference on Evolutionary Multi-Criterion Optimization. Berlin, Heidelberg: Springer Berlin Heidelberg, 2005.‏
>
> > On the assumption that  $\boldsymbol{\lambda}\in\mathbb{R}^n$ with $n\ll m,k$.
>
> Yes, it is correct. We assume that we have several hyperparameters to tune, typically up to 10 or 20, along with a large number of validation and calibration data points, ranging from a few hundred to several thousand, depending on the application. The number of objective functions (denoted by $c+1$ and typically no more than a few) is also significantly smaller compared to the data sizes.
>
> > On allowing repetition in the dataset created for the classification robustness experiment
>
> Yes, we allow for repetitions. Following your comment we added this clarification.
>
> > On specifying the notation in Eq. (4)
>
> We added explanations below Eq. (4).
>
> > On the confusion in $k$ and $\kappa$
>
> We fixed the notation.
>
> > On the definition of $\hat{\ell}^{\textrm{opt}}_1(\boldsymbol{\lambda})$
>
> We added the missing definition before Eq. (7).
>
> > On indexing error on Algorithm 1
>
> The index was fixed.

---

### Review · Reviewer_5hEm · 2024-07-19

**Summary Of Contributions:**

This paper considers the problem of hyperparameter optimization when there are constraints on quantities that must be estimated from data. For example, selecting the learning rate to minimize the classification error while bounding the worst case error on sensitive subgroups. The proposed approach GuideBO consists of two stages: 1) identify Pareto optimal configurations that lie in a region where the constraints are satisfied with high probability and 2) perform statistical testing on the identified configurations to find the one that optimizes the objective while rejecting the null hypothesis that at least one constraint is not satisfied. Experiments are done on a variety of settings like fairness, generative modeling, and network pruning. GuideBO outperforms existing baselines in terms of the objective value obtained.

**Audience:**

Yes

**Broader Impact Concerns:**

Since GuideBO is a general-purpose optimization algorithm, there are no direct societal impacts. No broader impact statement is included.

**Claims And Evidence:**

Yes

**Requested Changes:**

### Critical ###
- Empirically illustrate the budget size that is needed achieve an objective smaller than L, for a range of values of L.

### Other ###
- Discuss the possibility of theoretical guarantees.

### Questions ###
- How is the filtration for Pareto optimal configurations done? (Line 7 of Algorithm 1)

**Strengths And Weaknesses:**

### Strengths ###
- The proposed framework is applicable to a wide variety of problems.
- GuideBO is mathematically well-motivated and the approach is novel, as far as I know.
- The paper is clearly written.
- Experiments are done on a wide variety of problems and GuideBO is ranked first in terms of the objective value obtained in the majority of cases.
- Ablations are carried out on the strictness of the constraints and the size of the region where the proposed configurations lie.

### Weaknesses ###
- There are no theoretical guarantees on whether GuideBO is able to find the optimal configuration, even in the convex setting.
- The authors argue that GuideBO leads to more efficient statistical testing with fewer computations, but take an indirect approach in the experiments by fixing the budget of proposed configurations. A more direct and complete approach would be, for a set of objective values L, compare the N that is needed to achieve an objective smaller than L for each baseline.

---

> ### Author Response · Authors · 2024-08-01
> **Response**
>
> Thank you for your helpful and constructive review.
>
> > On illustrating empirically the budget size that is needed to achieve an objective smaller than L, for a range of values of L.
>
> Following your comment we added an experiment showing the value of the free objective as a function of the budget (Fig. E.2) and the budget that is required to achieve different objective levels  (Fig. E.3). We observe that in nearly all cases (15 out of 18 scenarios), GuideBo requires the smallest budget compared to the baselines, and in several cases, it achieves lower objective levels that are unattainable by the other baselines.
>
> To summarize, we demonstrate the efficiency of the proposed method compared to baselines in two ways:
> 1. For a **fixed budget**, we show that GuideBO achieves the lowest values for the free objective function compared to baselines (Figs. 5 and E.5).
> 2. For a **fixed level** of the objective function, we show that GuideBO requires the smallest budget to achieve that level (Fig. E.3).
>
>
> > On discussing the possibility of theoretical guarantees
>
> For theoretical analysis, we can compare GuideBo to the recovery of the full Pareto front. Note that our method is general and does not rely on the specific multi-objective optimization approach (though we propose practical implementation based on hypervolume improvement). For the analysis we assume that the multi objective optimization approach is able to recover the exact Pareto front in either the focused case (GuideBO) or in the general case (full Pareto front recovery). Further assuming a sufficient amount of validation data, this front coincides with the actual Pareto front defined by the expected objective values. As the number of validation points approaches infinity the region of interest defined in Eq. (8) shrinks. As a result, we will find configurations with expected (constrained) objective that approach $\alpha^\textrm{max}$. Conversely, if we aim to recover the full Pareto front with a fixed budget, the final set may not include a configuration that falls exactly on $\alpha^\textrm{max}$. Thus, testing the configurations chosen by GuideBO will result in better minimization of the free objective function compared to the full Pareto approach, thereby improving efficiency.
>
> In practice, we have finite validation data, leading to discrepancies between empirical objective values and expected values. A more in-depth analysis would examine the effect of sample complexity on the selection mechanism, specifying the probability for which GuideBO is expected to outperform a full Pareto front approach.
>
> > How is the filtration for Pareto optimal configurations done? (Line 7 of Algorithm 1)
>
> The filtration is done based on the criterion for Pareto optimality, specified in Eq. (2). For each configuration, we compare it against all other points in the set and confirm that it improves at least one objective value with respect to each other point.

---

> > ### Comment · Reviewer_5hEm · 2024-08-19
> >
> > Thank you for your response - I feel that my concerns have been adequately addressed.

---

### Review · Reviewer_aZFZ · 2024-07-22

**Summary Of Contributions:**

The submission 2917 introduces GuideBO, a novel method that integrates Bayesian optimization with rigorous risk control to find hyperparameter configurations meeting user-defined constraints while optimizing other designed metrics.
This algorithm first introduces the concept of a "region of interest" $R$ in the objective space $\mathbb{R}^{c+1}$, which significantly reduces the search space for candidate configurations.
Computationally, the method modifies the traditional BO procedure to focus on Pareto optimal configurations $\hat{\mathcal{P}}$ within the defined region of interest $\mathcal{C}_n$. After identifying candidate configurations, the method applies statistical testing tools (i.e. Learn then Test, LTT) to obtain the $(\alpha, \delta)$ -risk-controlling prediction.
Empirical experiments show that GuideBO effectively selects efficient and verified configurations under practical budget constraints, outperforming several baseline methods.

**Audience:**

Yes

**Broader Impact Concerns:**

This paper is a general algorithmic research that has no ethical implications and does not require an impact statement.

**Claims And Evidence:**

Yes

**Requested Changes:**

- (important) For changes on the simulations and proofs, please refer to the weaknesses above.
- The related work section is comprehensive but could benefit from a more structured approach. I suggest the authors discuss directly related works (Stanton et al., 2023; Salinas et al., 2023) in a separate paragraph rather than within a long domain description. Additionally, consider discussing some work on black-box model selection under multiple objectives (e.g. some work mentioned in [R2]), as the current "Additional related work" merely extends the MOO paragraph.
  - [R2] Karl, Florian, et al. "*Multi-objective hyperparameter optimization in machine learning—An overview.*" ACM Transactions on Evolutionary Learning and Optimization 3.4 (2023): 1-50.
- Eq. 1 falls into the typical definition of risk-controlling prediction. I suggest that the authors relate it to the concept and literature of $(\alpha, \delta)$- RCP.
In Eq. 3, the RHS should be $H(\hat{\mathcal{P}}; \mathbf{r})$ to be consistent with the definition provided below.
- The BO part in Sec. 4 could be quite confusing. What does $\|\mathcal{C}_n\|$ refer to here? Is it the same as $N_0$ defined earlier? The $\mathbf{l}$ is not defined. The authors also seem to use $\kappa,k, \mathbf{k},k_i$ inconsistently, lacking clear notation for each.
What do you mean by "After a new configuration is selected, it is evaluated and added to the pull", is that "pool"?
I'm familiar with BO and can infer the meanings, but for someone else, this might be somehow difficult to grab.
- Could the authors briefly summarize the limitations of their work?
- In Section 5.3, it should be $\mathbf{\lambda}^*=\arg\min_{\mathbf{\lambda}\in\mathcal{C}^{valid}}$. Similarly, in Algorithm C.1, the step should be $\mathbf{\lambda}^*\gets \arg\min_{\mathbf{\lambda}}$.
- Please fix typos:
  - page 1, paragraph 3, "Addressing from a different prospective": "prospective" should be "perspective"
  - page 2, "...and identify highly-preforming configurations": "highly-preforming" should be "highly-performing"
  - page 3, paragraph 2, "However, when used in model hyperparamaeter tuning ...": "hyperparamaeter" should be "hyperparameter"
  - Similarly, on page 3 & page 10, the word "hyperparmeter" should be "hyperparameter"
  - after Eq.5, "to be risk-condoling" should be "risk-controlling"
- Some observed incorrect sentences:
  - page 10: "(i) difference between the full-time prediction the early-time prediction" -> add "and"
  - Algorithm 1: "ParetoFront() filter Pareto optimal objective"  -> "filters"
  - Fig. 5 caption: "Presents ..." -> consider "Presentations of ..."
  - Conclusion: "lie the specified region"-> add ''in"; "The broad applicability and effectiveness is demonstrated" ->  "are"

**Strengths And Weaknesses:**

## Pros
- By providing a statistically validated method for model selection that ensures adherence to user-specified risk limits, this approach has an impact on the real-world deployment of learning models, where balancing multiple aspects under risk constraints could be crucial.
- The proposed algorithm steps are easy to understand and not overly complex in practical terms, making its potential for broader application quite feasible.
- The idea of deriving a region of interest guided by the testing goal is interesting to me. This technique of incorporating the testing goal into the optimization process could potentially attract attention from other relevant communities, such as researchers in the targeted learning field.
- The method leverages the efficiency of BO to explore the configuration space, coupled with a focused optimization strategy that avoids the computational burden of evaluating irrelevant configurations. It therefore inherits the advantages of BO.
- The authors' illustration figures are well-done and effectively convey the overarching concepts of the algorithm to the readers.

## Cons
- This method requires $X$ and $Y$ as input, thus it appears unsuitable for many semi-supervised (or unsupervised) learning problems in machine learning. What modifications to the algorithm and theory would be necessary to extend it to these different settings?
- Although the overall structure is clear, the current version lacks readability. Many sentences are presented in a wordy way, and some math symbols are not well-defined or self-consistent. Significant improvements in writing and presentation are needed, please refer to the subsequent comments for details.
- In the testing stage (i.e., the claimed risk-controlling), the authors primarily follow the proof techniques from LTT [R1]. This results in theoretical advancements in statistical efficiency being incremental. However, considering TMLR's criteria, this should not be regarded as a significant drawback.
  - [R1] Angelopoulos, Anastasios N., et al. "*Learn then test: Calibrating predictive algorithms to achieve risk control.*" arXiv preprint arXiv:2110.01052 (2021).
- I think some of the proofs in the appendix have issues, which might again weaken the contributions and solidness. Technically,
  1. The analysis simultaneously uses $P$ and $\mathbb{P}$ in one equation (like Eq. 26) without clarifying the distinction between the two. Do you mean to suggest that $P(\cdot)$ represents the vanilla probability, while $\mathbb{P}(\cdot)$ denotes the empirical probability (with finite sample)?
  2. Concentration inequalities require the values of the random variables to be valid. Thus, the authors assume the upper bound of the $\hat{\ell}^{cal}$ and $\hat{\ell}^{opt}$ does not exceed $1$. Is this assumption too strict and therefore meaningless in practice? If the loss exceeds this range, how should we scale the $\alpha^{max}$ and $R(\alpha,k,m,\delta,\gamma)$ to ensure our theoretical framework remains effective? Please provide a comment on this.
  3. Note that the $m$ is the number of random variables, not $n$. So I guess the Hoeffding’s inequality in Eq. 17 should be $P\left(\hat{\ell}^{cal}(\boldsymbol{\lambda}) -\ell(\boldsymbol{\lambda})\leq -t\right) \leq e^{-2mt^2},$ or $P\left(\hat{\ell}^{val}(\boldsymbol{\lambda}) -\ell(\boldsymbol{\lambda})\leq -t\right) \leq e^{-2kt^2}$.

  4. According to the  super-uniform $P(p\_{\boldsymbol{\lambda}}^{\mathrm{HF}}\leq u)\leq u$, shouldn't we obtain that $p\_{\boldsymbol{\lambda}}^{\mathrm{HF}}=e^{-2 n(\alpha-\hat{\ell}(\boldsymbol{\lambda}))^2}$? How do we transform the coeff $-2n$ in Eq. 24 into $-2m$, and what does $(\cdot)\_+$ represent? Shouldn't the square function always be non-negative?
  5. When deriving Eqs. 7 and 8 in the main body, one assumption the authors did not mention but implicitly used is that $\gamma=e^{-2kt^2}$ for $t>0$, with $0<\gamma\leq \frac{1}{2}$ (hand calculations, may not be true). This corresponds to the second line of Eq. 26 in the appendix. Please address this oversight and provide clarification on it.
  6. (Minor) In Eq. 27, it's unclear what the operation between the two scalars $\hat{\ell}\wedge \alpha$ represents, and the factor $e$ is also undefined. Meanwhile, the Eqs. 17, 18, and 19 should end with commas, not periods.
- Due to the lack of theoretical analysis from an optimization side (e.g., iteration bounds or sample complexity), the contributions of this paper rely heavily on empirical experiments (Sec. 6 and App. E). Unfortunately, most tasks in the experiments involve no more than $5$ hyperparameters in configuration, which represents a relatively small optimization space, potentially not convincing and undermining the paper's contributions.
  - I highly recommend that the authors include higher-dimensional model selection experiments to better highlight the advantages of the proposed algorithm. Additionally, plotting wall-clock time versus the number of hyperparameters $n$ would help illustrate the 'efficiency gap' between the new algorithm and the baselines in high-dimensional settings. Visualizing the region of interest (in low-dimensional toy cases) could further enhance the presentation of experiments.
- (Minor) The authors didn't submit the code for verification.

---

> ### Author Response · Authors · 2024-08-01
> **Response**
>
> Thank you for your helpful and constructive review.
>
> > On the suitability of the proposed method for semi-supervised/unsupervised learning problems
>
> The method is general and can be applied to different types of problems. For example, we demonstrate its application for generative models using variational autoencoders, which is an unsupervised model that requires only $X$ as input.
> The method can be applied in various settings, provided there is an access to validation and calibration data for performing optimization and testing. Following your comment we added a note on page 4, clarifying  that we use both $X$ and $Y$ in our problem formulation for the sake of generality. However, the method can be applied in other settings that do not require labels.
>
> > On the readability, wordy writing style and confusing notation
>
> We revised several sections to enhance readability by shortening and clarifying sentences. Additionally, we corrected problematic notations and included necessary clarifications.
>
> > On the theoretical advancements
>
> We would like to clarify that while our work builds upon well-established and rigorous testing tools, the primary contribution of our paper lies in its algorithmic and practical advancements. Searching through large and expensive configuration spaces is a complex task, especially when maintaining rigorous risk control. This complexity often poses challenges in both computational and statistical efficiency. Our core idea is to conduct a focused Bayesian Optimization (BO) procedure to enhance testing efficiency. The paper showcases diverse applications that benefit from the reliable selection of configurations adhering to user-defined constraints. The experimental results demonstrate that our method is versatile, consistent, and high-performing compared to other risk-controlling baselines.
>
> > On the use of $P$ and $\mathbb{P}$
>
> Thank you. The notation should be the same everywhere. We fixed it in the revised  manuscript.
>
> > On concentration inequalities for unbounded losses
>
> In the unbounded case, the Hoeffding (or Hoeffding-Bentkus inequality) no longer applies. Instead we can use p-values based on the central limit theorem, which are asymptotically valid. We added the definition of the central limit p-values in Eq. (31).
>
> >  On the number of samples in Hoeffding inequality
>
> We replaced n → m everywhere.
>
> > On the meaning of $(\cdot)_+$ in Hoeffding inequality.
>
> The notation means $(\cdot)_+=\max(\cdot,0)$.
>
> This operation is performed before the square function is computed. Recall that the p-value reflects the tail probability that a given an empirical loss is obtained under the hypothesis that the loss is greater than the limit $\alpha$ . This means that we are interested in cases in which the empirical loss is lower than $\alpha$. We derive the p-value from Eq. (24), by rearranging the inequality inside the probability for the case that $\hat{l}(\boldsymbol{\lambda})<\alpha$. Therefore we add $(\cdot)_+$ to ensure that $\hat{l}(\boldsymbol{\lambda})<\alpha$.
> If  $\hat{l}(\boldsymbol{\lambda})\geq\alpha$ then the power of the exponent is zero and the p-value becomes 1. Note that a probability of one trivially satisfies the super-uniformity requirement $\mathbb{P}(p\leq u)\leq u,\> \forall u \in[0,1]$.
>
> Following your comment we added clarifications for the p-value derivation.
>
> > On clarifying Eqs. (7), (8) and (28)
>
> Eq. (28) follows from Eqs.(20) and (21) that are obtained from Eqs.(18) and (19), respectively, by the transformation $u=e^{-2mt^2}$. Indeed $t>0$ and $u$ should be between $[0,0.5]$ since the probability on the left-hand side of Eq. (28) is non-negative and bounded by $1$.
>
> Following your comment, we added clarifications for the derivation of Eq.(28).
>
> > The meaning of $\hat{\ell}(\boldsymbol{\lambda})\wedge\alpha$ and $e$ + commas at the end of equations.
>
> The notation $x\wedge y$ means the maximum value of $x$ and $y$, and $e$ is the natural exponent. We added clarifications regarding the notation and fixed the commas.

---

> > ### Author Response · Authors · 2024-08-01
> > **Response - Part II**
> >
> > > On adding experiments with hyperparameters of higher dimension.
> >
> > Following your suggestion we conducted five additional experiments for the early-time classification task using various structured time-series datasets (see Table C.1), featuring large hyperparameter spaces ranging from 8 to 12 dimensions. The results, summarized in Figure E.5, demonstrate that GuideBO consistently outperforms the baselines in nearly all cases, ranking first in 20 out of 24 scenarios..
> >
> > It should be emphasized that the proposed method consists of two stages: (i) searching candidate configurations and (ii) multiple hypothesis testing. Note that only the first stage is affected by the dimensionality of the hyperparameter space. We utilize Bayesian optimization techniques, which are well-suited for optimizing large hyperparameter spaces. For exceptionally high-dimensional spaces, more advanced methods can be employed to enhance optimization efficiency [1]. In any case, our primary idea is to concentrate the search on the most relevant region in the objective space. This can be achieved in practice through different optimization mechanisms. We do not claim that  we better navigate through high-dimensional hyperparameter spaces compared to baselines. Our advantage lies in narrowing the search space to a confined region of interest within the objective space, allowing us to cover the area with fewer iterations. This focused search approach is particularly beneficial when dealing with high-dimensional hyperparameter spaces or when each configuration is costly to evaluate, such as in the case of retraining large models. In these scenarios, each optimization iteration is more expensive, making a targeted search with less iterations especially valuable.
> >
> > [1] Daulton, Samuel, et al. "Multi-objective Bayesian optimization over high-dimensional search spaces." Uncertainty in Artificial Intelligence. PMLR, 2022.
> >
> > > On illustrating the efficiency gap.
> >
> > To highlight the efficiency of the proposed method, we conducted a new experiment comparing the objective values obtained by the different methods as a function of the budget (see Fig. E2) and presenting the budget required to achieve specific objective levels (see Fig. E3). We see that GuideBO consistently performs well across all cases, whereas the other baselines show inconsistent performance, achieving better results in some instances and worse in others. Moreover, GuideBO requires fewer iterations to achieve the same or better results compared to the baselines.
> >
> > Note that the actual difference in running time will vary depending on the hyperparameter dimensionality and the model whose hyperparameters are being tuned; as we use larger architectures, the time gap increases. Consequently, we present results in terms of the budget difference, which is independent of application-specific factors and user-specific choices.
> >
> > To summarize, we demonstrate the efficiency of the proposed method compared to baselines in two ways:
> > 1. For a **fixed budget**, we show that GuideBO achieves the lowest values for the free objective function compared to baselines (Figs. 5 and E.5).
> > 2. For a **fixed level** of the objective function, we show that GuideBO requires the smallest budget to achieve that level (Fig. E.3).
> >
> > > On visualizing the region of interest
> >
> > A visualization of the region of interest is provided in Fig. E.8. for different tasks. It can be seen that the region of interest defines a confined section in the objective space that is smaller compared to the full Pareto front.
> >
> > > ⁠On the code
> >
> > We will publicly release the code after the paper is accepted for publication.
> >
> > >  On the related work section
> >
> > We reorganized the subsection discussing related work on BO and added two additional sections on: (i) black-box model selection under multiple objectives (on page 3) ; and (ii) incorporating user preferences in MOO (on page 19).
> >
> > > On adding a reference to Eq. (1) and correcting the notation in Eq. (3)
> >
> > We added the relevant reference and corrected the notation.
> >
> > > On the background of BO
> >
> > Following your comment, we have rephrased this section and added clarifications.

---

> > > ### Author Response · Authors · 2024-08-01
> > > **Response  - Part III**
> > >
> > > > On summarizing the limitations of this work.
> > >
> > > Following your comment we added a paragraph on limitations and future work on page 13:
> > >
> > > “Limitations and future work. While our proposed method establishes an efficient mechanism for selecting risk-controlling model configurations and improves upon previous work, it has some limitations. Multi-fidelity optimization is a popular method for hyperparameter optimization, where resources are allocated efficiently (Li et al., 2018; 2020). In this approach, additional resources (e.g., more epochs) are allocated to promising configurations that performed well with fewer resources, while configurations that showed poorer performance are discarded. Our current method cannot be directly applied in this setting. Moreover, the calibrated selection procedure requires splitting the data between the validation set used for selecting
> > > the subset of promising configurations, and the calibration set used for their verification. Although this is commonly done in other conformal prediction and risk control methods (Angelopoulos et al., 2021; Bai et al.,2022; Ringel et al., 2024), in many practical settings there is only limited available data. Another issue is that there might be a distribution shift between the validation/calibration data and test data, and thus the  selected configuration might not be risk-controlling with respect to shifted test data distributions (Gibbs & Candès, 2024; Zollo et al., 2024). These challenges should be addressed in future work.”
> > >
> > > >  On correcting min → argmin
> > >
> > > We corrected both Eq. (13) and line 10 in Algorithm C.1.
> > >
> > > >  Fixing typos and incorrect sentences
> > >
> > > Thank you for the careful reading. We fixed all mentioned problems.

---

> > > > ### Comment · Reviewer_aZFZ · 2024-08-04
> > > > **comment**
> > > >
> > > > Thank you for the rebuttal and clarifications, as well as the efforts in the additional hyperparameter experiments. I find the new experimental results and the revised proofs satisfactory; they appear to be technically sound. I also suggest the authors include the definition of $()_+$ in the main text, and appropriately introduce $\gamma$ before defining region $R$ (e.g., "there exists $\gamma \in(0,0.5]$ such that $\mathbb{P}(\cdot)\geq1-2\gamma$"). Overall, I believe this version of submission surpasses the acceptance threshold for TMLR.

---

> > > > > ### Author Response · Authors · 2024-08-05
> > > > > **Thank you for your response**
> > > > >
> > > > > Thank you for reviewing our updated manuscript and for your response. We added your suggested changes to the revised manuscript (after Eq. (5) and before Eq. (7)).

---

### Author Response · Authors · 2024-08-01
**General Response**

We would like to thank the reviewers for their time and effort in reviewing our paper. Their constructive comments helped us to improve the paper as reflected in the revised manuscript. Following their comments, we added additional experiments, made major changes to the paper’s writing, and clarified important aspects.

---

### Decision · Action_Editor_wvpL · 2024-09-16

**Recommendation:** Accept as is

**Comment:**

The paper proposes a method for constrained Bayesian optimization, where the goal is to optimize an unknown function (e.g. find optimal training hyperparameters for a machine learning model) while controlling risks (such as enforcing fairness constraints). The paper introduces the concept of a "region of interest" for satisfying the user constraints, and the proposed method aims to find the intersection of the Pareto front with this region of interest.

The paper is well-written, and provides sufficient evidence for the claims made. The reviewers engaged with the authors on the technical details of the proofs and experiments, and the current version of the paper includes the necessary clarifications. In particular, during the rebuttal stage the authors added new experiments with higher-dimensional optimization problems.

Overall, this is a strong high-quality paper, and it will be a valuable contribution to TMLR.

**Audience:**

The paper makes a contribution to the Bayesian optimization field, which is of interest to the TMLR community. It can also be of interest to the fairness and robustness communities.

**Claims And Evidence:**

The paper provides strong evidence for the claims made, with rigorous proofs and numerical experiments.